# Sim-LLM: Optimizing LLM Inference at the Edge through Inter-Task KV Reuse

**Ruikun Luo**[1234], **Changwei Gu**[1234], **Qiang He**[1234]*, **Feifei Chen**[5],
**Song Wu**[1234], **Hai Jin**[1234], **Yun Yang**[6]

[1]National Engineering Research Center for Big Data Technology and System
[2]Services Computing Technology and System Lab    [3]Cluster and Grid Computing Lab
[4]School of Computer Science and Technology, Huazhong University of Science and Technology
[5]Deakin University    [6]Swinburne University of Technology
{rkluo, gumorming, heqiang, wusong, hjin}@hust.edu.cn,
fefei.chen@deakin.edu.au, yyang@swin.edu.au

## Abstract

KV cache technology, by storing key-value pairs, helps reduce the computational overhead incurred by *large language models* (LLMs). It facilitates their deployment on resource-constrained edge computing nodes like edge servers. However, as the complexity and size of tasks increase, KV cache usage leads to substantial GPU memory consumption. Existing research has focused on mitigating KV cache memory usage through sequence length reduction, task-specific compression, and dynamic eviction policies. However, these methods are computationally expensive for resource-constrained edge computing nodes. To tackle this challenge, this paper presents Sim-LLM, a novel inference optimization mechanism that leverages task similarity to reduce KV cache memory consumption for LLMs. By caching KVs from processed tasks and reusing them for subsequent similar tasks during inference, Sim-LLM significantly reduces memory consumption while boosting system throughput and increasing maximum batch size, all with minimal accuracy degradation. Evaluated on both A40 and A100 GPUs, Sim-LLM achieves a system throughput improvement of up to 39.40% and a memory reduction of up to 34.65%, compared to state-of-the-art approaches. Our source code is available at https://github.com/CGCL-codes/SimLLM.

## 1   Introduction

The deployment of edge applications, such as autonomous vehicles and smart traffic management, has generated massive real-time data from numerous edge devices [1, 2]. In recent years, edge intelligence has further highlighted the advantages of edge computing in privacy preserving and latency reduction. *Large language models* (LLMs) can be deployed on edge servers close to users to improve their experiences [3]. However, as the number of tasks increases, the memory overhead required for processing these tasks increases rapidly, making it difficult to process on time. This problem becomes even more pronounced on edge servers, where resources are inherently constrained [2, 4].

The memory consumption of LLMs during inference primarily arises from the model parameters and the *key-value* (KV) cache [5, 6]. KV cache is a common technique for accelerating LLM inference. The core idea is to store previously computed key and value vectors from the attention mechanism for reuse in subsequent token generations. Since the size of KV cache grows linearly with the sequence length, batch size, and the number of model layers, it consumes a lot of GPU memory [7, 8].

---

*Corresponding author.

39th Conference on Neural Information Processing Systems (NeurIPS 2025).

A substantial body of research has attempted to mitigate the memory consumption of KV cache in LLMs, with most efforts aimed at reducing the sequence length stored in the cache to minimize its overall size. For example, LLMlingua [9] employs prompt compression techniques to reduce KV cache memory consumption. Similarly, Lm-infinite [10] introduces a method for compressing specific token spans into more compact representations to save memory. Additionally, the approach proposed in [11] employs a dynamic KV cache eviction policy to selectively retain only a small portion of the KV cache in memory. However, these methods were designed based on a layer-wise or token-wise perspective. When the number of tasks is large, their ability to reduce memory consumption remains limited, and they also incur significant computational overhead.

*Observation 1.* **There exists a broad similarity among LLM tasks in edge computing systems.**

This paper presents a novel approach named Sim-LLM to reduce the memory consumption of KV cache of edge nodes from a task-wise perspective. Its core idea is to reuse KV cache based on the similarity between previous and current inference tasks. Specifically, we have observed a considerable similarity among LLM tasks in edge computing systems through extensive experiments. Figure 1 shows the similarity proportions within and across seven widely used datasets for LLM inference in edge computing systems. Among them, REDDIT [12] contains the comments of 50 high-quality subreddits from the REDDIT PushShift data dumps (from 2006 to 2023), and we use the data collected after 2020 to ensure the up-to-dateness. MMChat [13] and LCCC [14] are conversation dialogues collected from Weibo, PTT,

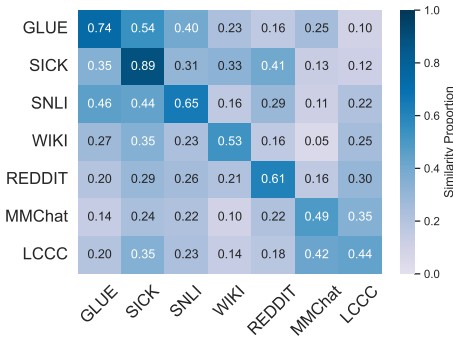

Figure 1: Similarity proportion in and between seven datasets

and Douban. As shown in Figure 1, there is a broad similarity among LLM tasks in edge computing systems (**Observation 1**). The reason is edge servers' limited service coverage and geographical distribution [2, 15]. The tasks handled by edge servers are more likely to reflect the hot events of the regions they serve. Thus, these tasks are likely to exhibit a high degree of similarity. Similar observation has been reported in previous studies [16, 17].

*Observation 2.* **KV caches generated by similar tasks also share similarity.**

Building upon the insights from **Observation 1**, we find that there is also a notable similarity among the KV caches generated by similar tasks (**Observation 2**). Figure 2a illustrates the similarity of top-layer key values across 40 similar inference tasks, with the majority concentrated above 0.7. To further quantify the degree of KV cache similarity, Figure 2b visualizes the similarity relationships among 10 similar inference tasks. Both Figure 2a and Figure 2b highlight the potential to reduce memory usage by exploiting the similarity between KV caches. Given the significant memory consumption of KV cache, as shown in Figure 3, **Obser-**

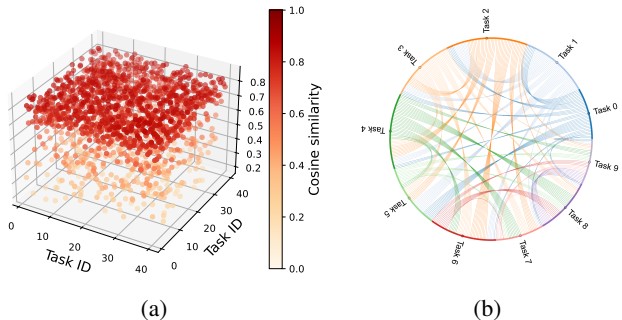

(a)            (b)

Figure 2: **(a)** Cosine similarity scores of top-layer Key values of 40 similar tasks. **(b)** Visualization of the similarity relationships between the top-layer key values of 10 similar tasks. In **(b)**, the wider the chord width of the same color, the more similar relationships the task has with other tasks.

**vation 2** serves as the basis for our proposal of a method that leverages similar KV caches to improve system throughput and reduce KV memory usage.

*Observation 3.* **The performance of the model is not significantly affected when reusing similar KVs.**

In Sim-LLM, each edge server retains the KV cache of previous inference tasks. By identifying the similarity between consecutive tasks, the KV cache from a previous task can be leveraged to accelerate

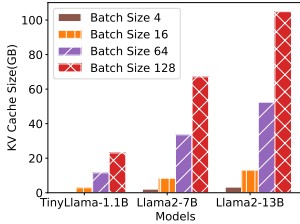

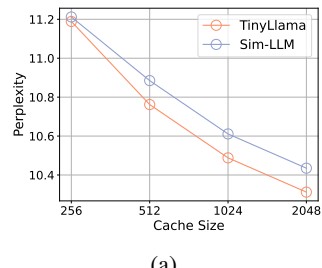

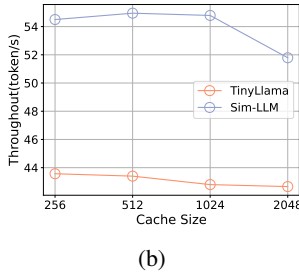

(a)             (b)

Figure 3: Memory usage of KV cache at token length 512 for different models

Figure 4: **(a)** Comparison of perplexity between TinyLlama [18] and our model w.r.t the cache size(number of processed tasks' sequence & KVs). **(b)** Comparison of throughput between TinyLlama and Sim-LLM w.r.t the cache size. The prompt length and generation length are both 2048 in (a) & (b).

subsequent inference tasks. Additionally, edge servers can communicate and exchange information about their respective tasks [19]. Similar tasks can be identified across them to provide another opportunity for inference acceleration. To ensure the feasibility of our approach, we conducted comparative experiments on model throughput and perplexity (PPL) using TinyLlama [18]. The results are presented in Figure 4, which further highlights that exploiting KV reuse can accelerate inference tasks and reduce memory usage without compromising model accuracy significantly (**Observation 3**). The main contributions of this paper are summarized as follows:

- To the best of our knowledge, Sim-LLM is the first work that leverages task similarity to accelerate LLM inference. We observe a common similarity among LLM inference tasks in edge computing systems and discover that the KV caches generated by similar tasks also exhibit similarity.
- Based on our observation, we propose Sim-LLM, an efficient inference optimization mechanism. Sim-LLM introduces a new method for the identification of similar inference tasks and a KV sharing mechanism across multiple edge servers to improve system utilization.
- Experimental results using PPL and various downstream benchmarks demonstrate that Sim-LLM reduces GPU memory usage by up to 34.65% and improves throughput by up to 39.40% against state-of-the-art approaches on average, without significantly affecting model accuracy.

## 2 Challenges and Motivation

The memory usage of KV cache increases linearly with the sequence length, batch size, and the number of model layers. As Figure 3 demonstrates, when the batch size is 128, the memory consumption of the KV cache alone is nearly three times the size of the model. In edge computing systems, **Observations 1 & 2** present opportunities to optimize KV cache memory consumption because many similar tasks can be received for LLMs during inference, which leads to the storage of numerous similar KV caches. Leveraging these observations, Sim-LLM aims to accelerate edge LLM inference. Specifically, instead of releasing all KV caches after inference, Sim-LLM retains a portion of the KV cache and shares it with subsequent similar tasks. Similar KV caches can also be shared across edge servers, where communication and task information exchange among servers allow for identifying similar tasks already completed in an edge computing system. In such cases, edge servers can offload similar tasks to one another to enable inference acceleration. However, Sim-LLM must tackle three main challenges.

- **Adaptive similar tasks identification**: How to effectively identify similarities between tasks while minimizing the overhead introduced by the identification process.
- **Efficient handling of unique tasks**: Given that not all tasks are similar, how to handle unique tasks efficiently without compromising system performance.
- **Reuse of inter-task KV across edge servers**: How to identify similar tasks across multiple edge servers, and upon detecting similarities, how to determine the optimal data to be transmitted for efficient task offloading.

**To address the first challenge**, extensive research is conducted, revealing that cosine similarity provides significant advantages in identifying semantic similarities between tasks, particularly in the context of textual data. As the number of KV to be stored increases, the overhead incurred

from traversing the cache to check for similar tasks may diminish the benefits of utilizing similar tasks to accelerate inference. To mitigate this issue, we adopt a *Locality-Sensitive Hashing* (LSH) mapping approach [20]. When the current task being processed is mapped to the same LSH bucket as a previously inferred task, there is a high likelihood that they share semantic similarity. However, due to the potential distortion introduced by LSH, we implement an adaptive mapping strategy. Specifically, when the batch size of request sequences is small, we compare the current sequence and previously stored sequences' cosine similarity. For large batch sizes, we employ LSH to identify similar sequences. Since LSH may incur a distortion issue, where highly similar previous tasks are mapped to different hash values or two dissimilar tasks may be mapped to the same bucket, degrading the acceleration performance, Sim-LLM merges KVs within the LSH bucket before being utilized in the inference process to address this issue.

**To address the second challenge**, tasks that have similar tasks can leverage the top-layer KV from these corresponding similar tasks to accelerate inference. This approach is inspired by interpreting the Transformer's stacked layer structure as an iterative process that refines token representations [21]. In this context, the representation at the top layer is considered the most informative. Thus, prioritizing the top layer when evaluating task similarity is a proper choice. However, tasks without similar counterparts requires recomputing KV from scratch during inference, resulting in a significant slowdown compared to requests that can leverage previous KVs. Inspired by inter-layer KV sharing methods like YOCO [22] and CLA [23], only the KV for certain layers at the bottom and top of the model are cached. By excluding the need to compute KV for intermediate layers, and omitting the associated weight parameters $W_K$ and $W_V$, the overall computational and parameter overheads are minimized. Furthermore, this approach retains the potential for integration with intra-layer KV sharing methods such as MQA and GQA techniques, and can benefit from dataflow optimizations like FlashAttention [5].

**To address the third challenge**, Sim-LLM combines LSH mapping with prototype learning [24]. While LSH mapping is more compatible with and better suited to the task processing mechanism for the single-node scenario discussed in Section 3.1, it can introduce significant query overhead when searching for similar tasks repeatedly across multiple edge servers. Inspired by the concept of prototype learning [24], to overcome this, Sim-LLM extracts and merges task features processed by each edge server to generate a task prototype. Each edge server is required to maintain a global feature table. By comparing the features of an incoming task with those stored in the table, the most likely edge server with similar tasks can be identified. In cases where an edge server does not store any similar tasks, it may optionally seek assistance from its neighboring servers before proceeding with the normal inference process. This approach accelerates task identification and enhances offloading efficiency, improving the overall performance and scalability of inference in a distributed edge computing system.

## 3 Sim-LLM

### 3.1 Overview

Sim-LLM aims to improve the efficiency of LLM inference while reducing KV cache consumption in edge computing systems, which consists of two distinct processes: one for single-node scenarios and another for multi-node scenarios.

In single-node scenarios, Sim-LLM leverages the similarity between tasks. After processing each batch of requests, the embeddings and top-layer KV of the processed requests are cached. The component responsible for managing this data is referred to as the *KV_Manager*. During task processing, the system first checks whether the cached tasks in the KV_Manager are similar to the task currently being processed. If a match is found, the system utilizes the cached top-layer KV to accelerate inference; otherwise, it performs a normal inference.

In multi-node scenarios, each edge server processes tasks in the same manner as in the single-node scenario. To maximize resource utilization in the system, task similarity is fully exploited. Edge servers communicate over the network to determine whether they have cached similar tasks to the one currently being processed. If similar tasks are found on one of the edge servers in the system, the task request is offloaded to that server for processing with the cached similar KV.

Figure 5 illustrates the overall workflow of Sim-LLM, which will be described thoroughly in Section 3.2 and Section 3.3. In the preprocessing phase, inference tasks are tokenized to word embeddings

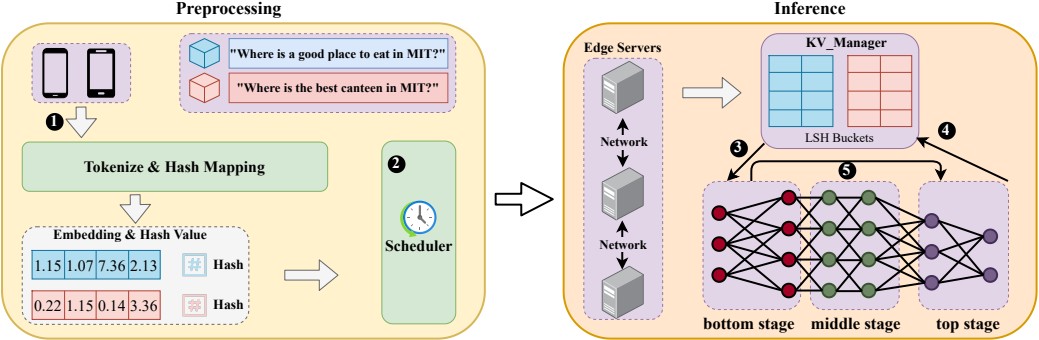

Figure 5: Sim-LLM overview

before the Scheduler assigns them to edge servers. During the inference phase, KV replacement and sharing are employed throughout the three-stage model inference process on edge servers.

## 3.2 Semantic Similarity Identification

**How to define whether an inference task is similar to previous tasks?** One commonly used approach to measure text similarity is Jaccard similarity. However, Jaccard similarity has a notable limitation. It ignores variations in text lengths. For example, the two sentences "I love you!" and "I love you! I love you! I love you!" would yield a similarity score of 1, implying that they are identical, despite having different semantic meanings due to the repetition in the second sentence. Metrics such as Euclidean distance, Levenshtein distance, and Jaro-Winkler distance are also strongly influenced by text length when measuring similarity, making them unsuitable for evaluating tasks when the text length is unknown. Therefore, the semantic-based method of cosine similarity is preferred. This method is widely used in word vector models (e.g., Word2Vec [25], GloVe [26]) and semantic models (e.g., BERT [27]). Regarding the definition task similarity, Word2Vec and GloVe consider a cosine similarity threshold of 0.6, Sentence-BERT and SemEval use 0.7. Through evaluation of the threshold's impact both on generation speed and model accuracy, Sim-LLM employs a stringent threshold to offset performance degradation after KV reusing while maintaining inference efficiency, considering two tasks to be similar when their cosine similarity exceeds 0.8. The detailed results can be found in Appendix 4.5.

**How to recognize the semantic similarity between tasks?** When employing an exhaustive search approach to compare the similarity between incoming tasks and cached tasks, Sim-LLM outperforms the state-of-the-art approaches in various downstream tasks. However, this advantage is limited to small batch sizes. As batch size increases, the performance advantage of Sim-LLM diminishes. This performance decline is caused by the cumulative computational overhead introduced by the exhaustive search for similarity calculation, which reduces the inference speed. To address this issue, Sim-LLM employs LSH mapping. During the task caching process, it stores the task's corresponding embedding, the top-layer KV, as well as the hash value calculated from the word embedding. By transforming the word embedding of an incoming task into a hash value and mapping it to an LSH bucket, Sim-LLM can efficiently identify cached tasks similar to the current task. By reducing high-dimensional data to lower-dimensional representations, LSH can significantly accelerate task similarity identification.

## 3.3 KV_Manager

KV_Manager stores the embeddings and top-layer KVs of previous tasks so that Sim-LLM can match similar tasks and utilize similar KVs to accelerate inference. As discussed in Section 2, it leverages similar top-layer KVs to replace the conventional KV generation. Moreover, it has been observed that Transformers focus on syntactic information in the lower layers and semantic information in the higher layers [28], in addition to the informative characteristic of top-layer KV. Thus, it makes sense to consider the top-layer KV for replacement between tasks sharing semantic similarity. Thus, there is no need to cache the KV from other layers. Consequently, Sim-LLM caches only the keys and values from a single layer, unlike a typical Transformer model that caches those from multiple layers. This significantly reduces memory consumption without introducing additional computational overhead during inference.

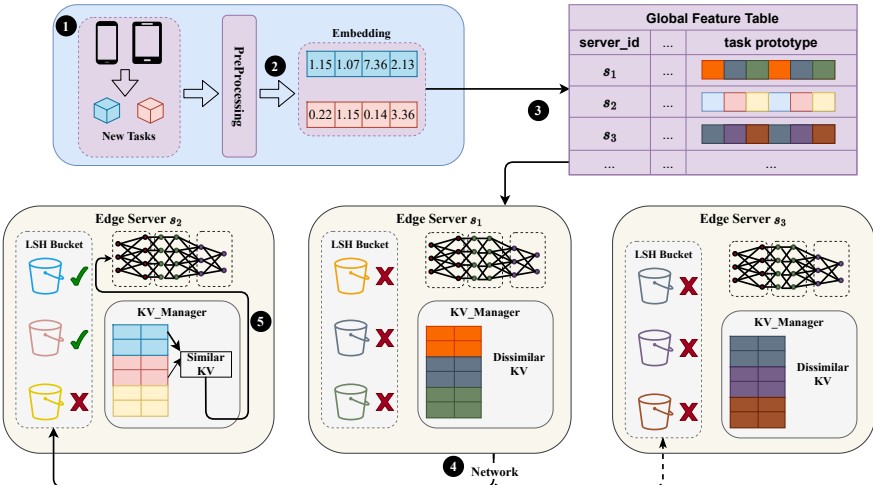

Figure 6: Multi-nodes share tasks with LSH and task prototype

Although the KV_Manager introduces extra memory requirements for storing the embeddings and top-layer KVs of previous tasks (which will be discussed in Section 4.2), this extra memory usage is substantially lower than the memory overhead associated with computing and storing KVs of all tasks. Furthermore, KV_Manager facilitates KV sharing across multiple edge nodes to further reduce the additional memory overhead, as discussed in Section 3.4.

As shown in Figure 5, the KV replacement process within KV_Manager consists of five steps. ❶ When a batch of tasks arrives, a tokenizer first tokenizes these tasks to obtain their embedding vectors. Then these embeddings are mapped to corresponding hash values. ❷ The Scheduler distributes these tasks' embeddings and hash values to neighboring edge servers to identify similar tasks. ❸ KV_Manager stores the embedding vectors, hash values, and top-layer KV of tasks that have been inferred. Before inference begins, the tasks are mapped to the LSH buckets in KV_Manager to facilitate the search for similar tasks. If similar tasks are found, the top-layer KV stored in the KV_Manager is passed to the first layer of the LLM (bottom stage). This operation accelerates the prefilling phase by eliminating the need to compute the KV from scratch at the attention module. ❹ When the prefilling phase (top stage) for the first new token is about to complete (new tasks' top-layer KV have been computed), KV_Manager stores the top-layer KV as well as the embedding of the current task and applies an eviction algorithm to remove the KV of outdated tasks. ❺ Since the top-layer KV is utilized, the inference process of lower layers can be skipped. Inference can be performed directly at the final layer of the model to produce the output (top stage), thereby skipping the entire prefilling phase.

For matched tasks, the queries from all layers are paired solely with the top-layer KV pair of similar previous tasks. This eliminates the need to cache or compute the KV for all layers, saving memory consumption and computational overhead. Furthermore, by using the top-layer KV to replace the KV generation process for the current task across all layers, Sim-LLM bypasses intermediate layers and directly generates the output token from the top layer. However, this method assumes that every task within the current batch has a corresponding similar task with an available top-layer KV, which is not always true. To address this, when more than half of the tasks in a batch are matched, those without a corresponding top-layer KV are temporarily stored for the next batch. This ensures that the remaining tasks can bypass the initial computation process, thereby enhancing the overall inference efficiency. For unmatched tasks, the goal is to accelerate their inference process by optimizing the model configuration. Sim-LLM retains the KV from the bottom three layers (bottom stage) and the top three layers (top stage) in a 'sandwich' structure.

KV_Manager continuously maintains metadata for newly processed tasks. As cached tasks accumulate, the memory footprint grows commensurately. When the cache reaches the cache size, eviction is triggered to reclaim space for incoming tasks and to prevent memory exhaustion. Accordingly, KV_Manager adopts the *Least-Recently-Used* (LRU) eviction policy that preferentially preserves frequently reused task KVs while removing those accessed least recently. This design aligns with the central premise of this work: reusing as many similar KVs as possible to accelerate inference.

## 3.4 Across Nodes KV Sharing Mechanism

In Sim-LLM, edge servers can exchange their cached tasks to accelerate the overall inference speed. To further improve memory utilization, Sim-LLM employs a across nodes KV sharing mechanism.

As shown in Figure 6, let us assume three edge servers in the system. Sim-LLM utilizes LSH to map the current task from edge server $s_1$ into the LSH bucket corresponding to edge server $s_2$ and $s_3$. The process consists of these steps: ❶ Users submit their tasks to edge servers. ❷ The new tasks are tokenized through a preprocessing module. ❸ The scheduler assigns the tokenized embedding to $s_1$ after comparing the user tasks' feature with the servers' task prototype in the global feature table. $s_1$ then begins searching for similar tasks within its LSH bucket. ❹ If no similar tasks are found, $s_1$ attempts to transmit the new embeddings to its neighbor servers. ❺ If a neighbor server identifies a similar task, it will take over the remaining inference process.

When an edge server $s_i$ attempts to find tasks similar to the current task among its neighbor servers, it begins the inference process of the current task itself. Its neighbors search for similar tasks simultaneously. When a similar task is found on a neighbor edge server, that server sends a signal to $s_i$ to halt its inference and accelerates the task using the similar task. However, this approach introduces a potential issue: $s_i$ may be interrupted just as its inference nears completion. This incurs resource waste. A straightforward solution to this problem is to limit the number of hops for the queries. Inspired by the concept of prototype learning [24], Sim-LLM extracts features from inference tasks on each edge server to identify task features they often handle. Each edge server maintains a global feature table that contains a prototype for each server. By comparing the features of incoming tasks with those stored in the table, it can identify the edge server that is most likely to have similar tasks. With this method, Sim-LLM can locate a server with similar tasks with only one query.

Despite the across node KV sharing mechanism improves the inference performance, it also incurs communication overhead due to information exchange between edge servers. To address this issue, when similar tasks are identified on another edge server, only the sequence or the corresponding word embeddings are transmitted to the identified edge server, rather than transmitting the KV cache of the similar task to the current processing task from the identified server back to the original server. However, if the identified edge server is occupied with processing other tasks, it must transfer the stored KV cache back to the original edge server for accelerating inference.

## 4 Experiments

### 4.1 Experimental Settings

**Models**. To evaluate the performance of Sim-LLM, experiments are conducted on two widely-used English LLMs, i.e., TinyLlama-1.1B [18], Llama2-7B and 13B [29], as well as a bilingual LLM, i.e., InternLM2-7B [30], which supports both Chinese and English. For the experiments, we utilize the chat versions of Llama2-7B, InternLM2-7B, and Llama2-13B.

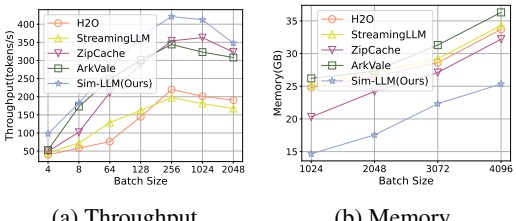

(a) Throughput      (b) Memory

Figure 7: Comparison of throughput and memory consumption between Sim-LLM and baselines on Llama-7B

**Benchmarks**. The experiment is conducted in the OpenCompass evaluation framework contributors [31] and the lm-eval-harness framework [32]. The evaluation involves five key aspects, i.e., reasoning, language, knowledge, examination, and understanding, with corresponding benchmarks: (1) **Reasoning:** CMNLI [33], HellaSwag (HeSw) [34], PIQA [35]; (2) **Language:** CHID [36], WSC [37]; (3) **Knowledge:** CommonSenseQA (CSQA) [38], BoolQ [28]; (4) **Examination:** MMLU [39], CMMLU [40]; (5) **Understanding:** Race-High/Middle (H/M) [41], XSum [42], C3 [43].

The evaluation is conducted with the official scripts from OpenCompass, employing a zero-shot approach without additional training. Two evaluation modes are utilized: perplexity (PPL) and generation (GEN)[2]. The GEN mode is used for the CHID and XSum benchmarks, while both PPL ($WSC_P$) and GEN ($WSC_G$) modes are used for the WSC benchmark. The remaining benchmarks are evaluated in the PPL mode. OpenCompass subsequently converts the evaluation results for each benchmark into a score, with higher scores indicating better performance.

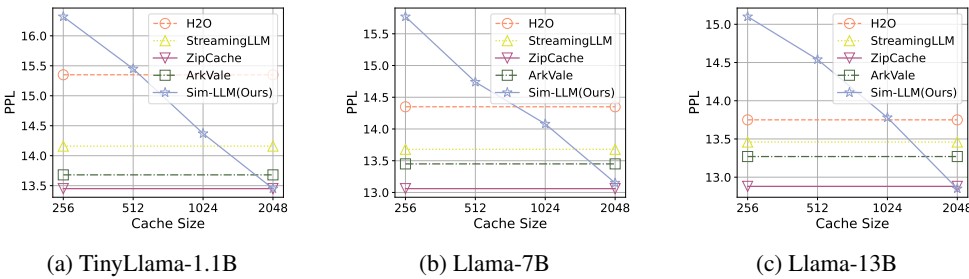

(a) TinyLlama-1.1B      (b) Llama-7B      (c) Llama-13B

Figure 8: PPL on Wikipedia dataset [45] at different cache sizes

**Experiment Environment**. The performance of Sim-LLM under single-node scenarios is evaluated on a server with a single Nvidia A100 80GB GPU. Four physical machines, each equipped with four Nvidia A40 40GB GPUs, are used as edge servers to evaluate the performance across edge nodes. All experiments related to the PPL evaluation are conducted on a 10M subset of the development set from SlimPajama [44] and the Wikipedia dataset [45].

## 4.2 Performance Results

Experiments are conducted on each dataset, and the average score across all tasks for each aspect is reported.

The baselines include the standard Transformer models as well as the state-of-the-art KV cache eviction and intra-layer compression methods, including StreamingLLM [46], H2O [47], ZipCache [48], and ArkVale [49]. Their details can be found in Appendix A.

We first evaluate the generation performance of Sim-LLM. Figure 7a and Figure 7b compare the throughput and memory consumption of the baselines and Sim-LLM on servers with A40 GPUs with varied batch sizes. Both the prompt length and generation length are set to 2,048. Sim-LLM outperforms all baselines across all tested batch sizes, achieving a speedup of up to 39.40% (33.04% on average) and memory reduction of up to 34.65% (30.05% on average). Please note that the largest batch size does not always yield the maximum system throughput. Notably, the increase in system throughput is not solely attributed to the increased batch size. By substituting the KV computation during the prefilling phase with the replacement of top-layer KV of similar tasks, Sim-LLM effectively eliminates redundant calculations, leading to both higher throughput and lower memory consumption. Additionally, we observe that as the batch size exceeds 256, system throughput no longer increases and even decreases when the batch size reaches 1,024. This suggests that the model transitions from memory-bound to compute-bound [5].

To validate the generation quality of Sim-LLM, we evaluate its PPL on various downstream benchmarks. Figure 8 demonstrates the PPL of both the baselines and Sim-LLM for the Llama series under different cache sizes. The PPL achieved by Sim-LLM is better when the cache size is larger. Since Sim-LLM directly utilizes the stored KVs, the computational complexity during the generation process is reduced, resulting in fewer inference steps and faster convergence to a lower PPL. When a similar task is identified, the corresponding KV of previous tasks serves as a supplement, providing context

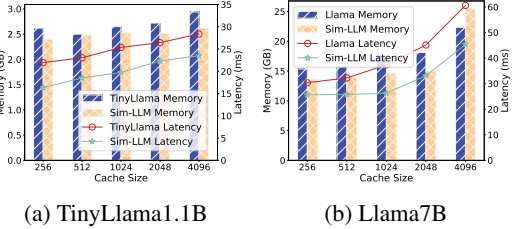

(a) TinyLlama1.1B      (b) Llama7B

Figure 9: Comparison of latency per token and memory consumption of (Tiny)Llama and our model w.r.t. different cache sizes

from similar scenarios that Sim-LLM has previously encountered. This enables Sim-LLM to generate outputs with greater certainty, reducing its reliance on unknown contexts. Essentially, Sim-LLM becomes more familiar with task types over time, leading to a decrease in PPL. Eviction-based methods, H2O and StreamingLLM, exhibit higher PPL compared to ZipCache, primarily due to their information loss upon token eviction. In contrast, the quantization-based method ZipCache preserves full data integrity, thereby achieving better performance. Furthermore, we evaluate zero-shot performance on the benchmarks mentioned in Section 4.1, using the lm-eval-harness framework [32]. Table 2 shows the average score across all tasks in Appendix B.4, demonstrating that Sim-LLM does

not experience significant performance drops in any specific area. The PPL results in Figure 8 and downstream benchmarks scores in Table 2 illustrate that Sim-LLM effectively preserves both overall and task-specific performance, maintaining reliable output quality.

To evaluate the impact of cache size on Sim-LLM, we conducted experiments using various cache size configurations. Figure 9 illustrates the latency and memory usage of Llama models with varying cache sizes incurred by Sim-LLM. Sim-LLM significantly reduces model inference latency and effectively decreases memory usage across most cache size configurations. As shown in Figure 9b, only when the cache size reaches 4096, the memory usage of Sim-LLM becomes larger than that of Llama. It results from the stored KVs and embeddings from previous tasks, offsetting the benefits of reducing memory usage by reusing similar KVs.

Usually, the sequence length does not exceed the maximum input lengths the models are trained on. To highlight the capability of Sim-LLM to support models with larger sequence lengths, we also evaluate this situation. Table 3 in Appendix B.3 compares the maximum batch sizes and throughput of standard Llama models and Sim-LLM across two types of GPUs. These results confirm it and show Sim-LLM's capacity to handle significantly larger batch sizes and achieve higher system throughput than all baselines across all settings.

### 4.3 Impact of Source Layer Used for KV Reusing

In this section, we evaluate the PPL and accuracy of downstream tasks on Llama-7B (total 32 layers) when choosing different layers as a similar KV source. The position of the source layer is from *top, middle, bottom*. We select several layers from these positions respectively, e.g., layer 3 to 5 for *bottom*, 14 to 16 for *middle*, and 30 to 32 for *top*. The cache size is set to 512. Figure 10 substantiates that the choice of top-layer KV as source is reasonable. Detailed results of downstream tasks are shown in Appendix B.4. It can be seen that *top* layer outperforms *bottom* and *middle* layer in both PPL and downstream task accuracy when performing KV sharing. Additionally, it can be observed that as the position of the source layer increases, the model's performance improves, gradually approaching that of the standard model. In addition to the reasons discussed in Section 2 and Section 3.3, we analyze that this phenomenon arises because, at lower layers, the KV lacks sufficient semantic information, which is inadequate to enable the model to achieve reliable performance.

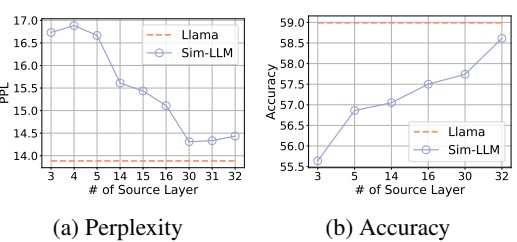

(a) Perplexity      (b) Accuracy

Figure 10: Perplexity on the SlimPajama dataset and average downstream task results of Llama-7B

### 4.4 Impact of Similar Task Proportion

To evaluate the impact of similarity proportion on Sim-LLM, we select tasks from REDDIT [12], MMChat [13], LCCC [14], WIKI [45], GLUE [50], SICK [51] and SNLI [52] to create task sets with varying proportions of similarity, where two tasks are considered similar if their cosine similarity exceeds 0.8. Evaluations are conducted on Llama-7B using the specified task sets, with a fixed prompt length of 512 and a generation length of 4,096. When the similarity proportion is zero, Sim-LLM performs inference in the same manner as standard Transformers. As demonstrated in Figure 11, with the similarity proportion in the task set increasing, Sim-LLM can more effectively leverage the reuse of KVs from similar tasks, thereby accelerating inference and improving throughput, although this may result in a slight degradation in accuracy.

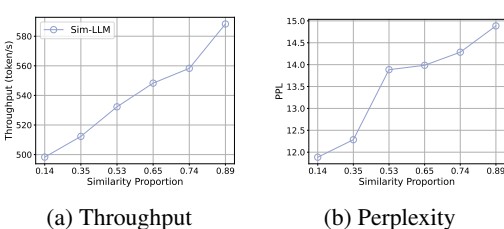

(a) Throughput      (b) Perplexity

Figure 11: Results of task sets with different proportions of similarity on Llama-7B

### 4.5 Impact of Cosine Similarity Threshold

Since Sim-LLM accelerates inference by exploiting task similarity, we investigate the impact of the similarity threshold on performance. Specifically, we determine the optimal cosine similarity threshold for KV reuse by evaluating generation speed, perplexity, and accuracy on downstream tasks. Figure 12 presents the results. In Figure 12a, the prompt length and the generation length are set to

512 and 4,096, respectively. When the threshold of cosine similarity is low, throughput performance exceeds that of a higher threshold. This is because most incoming tasks can easily identify "similar" tasks' KVs for reuse, introducing minimal identification overhead and accelerating the inference process. However, such an advantage comes at the expense of model accuracy, as illustrated in Figure 12b and Figure 12c. When the threshold reaches 0.9, there is a significant drop in generation speed due to fewer tasks being able to reuse KVs, resulting in many tasks performing inference as in the standard Transformer model. Therefore, as a trade-off between generation speed and model accuracy, a threshold of 0.8 is preferred.

## 5   Related Work

Extensive research has been dedicated to reducing the memory consumption of the KV cache for efficient inference of LLMs. Existing studies can be categorized into three main approaches.

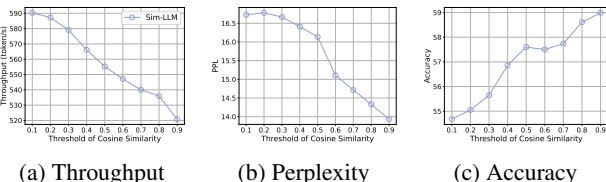

|             (a) Throughput             (b) Perplexity             (c) Accuracy |

**Memory Space Optimization.** KV caches can be scheduled properly to maximize resource utilization in edge computing systems. Infinite-LLM [53] partitions the KV cache into smaller, more manageable units

Figure 12: Results of different thresholds of cosine similarity on Llama-7B. Evaluation conducted on SlimPajama [44] dataset. The downstream tasks used in (c) are CMNLI [33], HellaSwag [34], and XSum [42].

for independent processing and management across the global memory scheduler in a distributed system. LoongServe [54] dynamically assigns the prefilling and decoding phases of tasks to instance groups, which can scale up or down in response to changing load demands, without incurring KV migration overhead.

**Memory Usage Optimization.** KVs can also be selectively retained to save on system resources [46, 47, 55, 56]. Scissorhands [56] and H2O [47] preserve only the crucial components of the KV cache based on attention scores. StreamingLLM [46] retains only the recent context window and a few initial tokens as an attention sink, and discards the rest of the past context. SnapKV [55] focuses on pruning tokens in the input prompt in response to increasing input lengths.

**Other KV Optimization Methods.** Other methods for KV optimization include quantization [8, 48, 57, 58], eviction [49, 59], and merging [7]. Quantization is a representative method for KV cache compression. It reduces the KV cache size from 16-bit to 4-bit or even lower. CacheGen [8] is particularly relevant as it combines KV offloading with quantization, mitigating KV transfer overhead by applying quantization-based compression algorithms. Other methods have also been proposed to retain only the KV cache of important tokens while discarding others to save GPU memory [46, 47, 55]. However, since the importance of tokens often changes during the decoding process, discarded tokens may become crucial for further computation [59], which potentially leads to accuracy loss. Quest [59] mitigates this issue by dynamically selecting a small portion of critical KV cache for attention computation for each query token, while retaining all KV cache.

## 6   Conclusion and Future Work

This paper present Sim-LLM, a novel method for reducing memory consumption and improving throughput for LLM inference in edge computing systems. It reduces the key-value computation overhead by reusing the KV calculated for previous tasks similar to current tasks. Extensive experiments demonstrate that Sim-LLM achieves reductions of up to 34.65% in memory usage and improvements of up to 39.40% in throughput, with negligible performance degradation in accuracy.

The primary limitation of Sim-LLM lies in the necessity of storing the top-layer KV pairs, as well as corresponding embeddings of processed tasks. Although Sim-LLM performs effectively across a wide range of cache size configurations, the storage required for saving additional KVs and embeddings may exceed the benefits gained from reducing the KV cache through the use of similar tasks. Furthermore, since the top-layer KVs computed by the model are essential, the KV shape across all edge servers must remain consistent. Consequently, the models deployed across edge servers must be identical. Our future work will explore how to share KVs for heterogeneous models.

## Acknowledgments

We sincerely thank the AC and reviewers for their constructive and valuable feedback. This research was supported in part by the National Science Foundation of China (NSFC) under grant No. 62402188, the China Postdoctoral Science Foundation under Grant No. 2024M761017, 2025T180424, the Postdoctoral Fellowship Program of CPSF under Grant No. GZC20240542, the Postdoctoral Project of Hubei Province under Grant Number 2024HBBHCXA026, and the HUST Kunpeng Ascend Sci-Edu Innovation and Incubation Center.

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

# Appendix

## A    Baselines

The performance of Sim-LLM is compared against four baselines.

**StreamingLLM** [46]: StreamingLLM is a long-context streaming processing approach designed for LLMs. Its core concept centers on stabilizing generation performance by retaining recent tokens and initial "attention sinks." It maintains a fixed-size cache window that preserves only the KV of recent tokens and a small number of initial tokens. Its primary limitation lies in the inherited constraint of pre-trained context window sizes, which prevents enhancement of the model's long-term memory capacity.

**H2O** [47]: With the core idea rooted in the optimization of computational efficiency during the generation process through the introduction of the "Heavy-Hitter Oracle" mechanism, H2O reduces unnecessary computations by identifying and prioritizing the most probable output tokens (i.e., heavy-hitter options), thereby enhancing the speed and efficiency of inference.

**ZipCache** [48]: The main concept of ZipCache is to precisely identify the tokens that have a significant impact on model generation, and to prioritize retaining information about these key tokens when compressing the KV cache. It introduces a channel-separable, per-token quantization strategy to effectively reduce the memory overhead of quantization parameters.

**ArkVale** [49]: ArkVale organizes the tokens in the KV cache into pages and maintains summary information for these pages. Before each attention computation, ArkVale evaluates the importance of each page based on the current query and the summary information of the cached pages. By dynamically evicting and recalling KV pages, it addresses the issue of token importance changing during decoding steps, without requiring modifications to the model architecture or fine-tuning. ArkVale [49] is the state-of-the-art block-level dynamic sparse attention within the vLLM.

## B    Additional Experimental Results

### B.1    Scalability and Latency of Sim-LLM Under Variable Edge Workloads

To evaluate Sim-LLM under heterogeneous edge workloads, we consider two task distributions: **Poisson distribution** and **Power-law distribution**. The Poisson case approximates a uniform workload, whereas the power-law case captures bursty, highly variable demand. As summarized in Table 1, under bursty conditions Sim-LLM remains scalable and does not introduce appreciable additional latency relative to the uniform-load scenario. We attribute

Table 1: Latency performance of Sim-LLM under different cache sizes for bursty and uniform workload patterns

| Cache Size | Poisson Latency (ms) | Power-law Latency (ms) |
| --- | --- | --- |
| 256 | 25.66 | 27.68 |
| 512 | 25.75 | 28.51 |
| 1024 | 26.31 | 29.34 |
| 2048 | 33.13 | 35.88 |
| 4096 | 45.35 | 47.72 |

this robustness to the LRU eviction policy: bursty traffic induces many tasks with overlapping or identical features, and LRU adapts by retaining frequently reused task KVs, allowing a large fraction of requests to hit reusable KVs. This improves cache locality and accelerates inference, rendering LRU particularly well suited to bursty workloads.

### B.2    Zero-shot Accuracy on Benchmarks

In this section, we conducted zero-shot accuracy evaluation on benchmarks discussed in Section 4.1, using the TinyLlama-1.1B, Llama2-7B, and Llama2-13B with official scripts from the lm-eval-harness framework [32]. The results in Table 2 show Sim-LLM is comparable to the standard transformers and the state-of-the-art methods in terms of accuracy on downstream tasks.

### B.3    Maximum Generation Batch Size and Throughput

In this section, we conducted generation performance evaluation with different prompt and generation lengths, using the TinyLlama-1.1B, Llama2-7B, and Llama2-13B with official scripts from the OpenCompass framework [31]. The results in Table 3 show that Sim-LLM outperforms the state-of-the-art methods.

Table 2: Zero-shot accuracy on different benchmarks w.r.t different models

| Model | Reasoning | Language | Knowledge | Examination | Understanding | Avg |
|---|---|---|---|---|---|---|
| TinyLlama-1.1B | 44.58 | 30.2 | 50.99 | 46.38 | 25.00 | 39.43 |
| H2O | 43.68 | 28.52 | 43.90 | 20.94 | 25.37 | 32.48 |
| StreamingLLM | 45.66 | 28.22 | 45.11 | 22.32 | 24.36 | 33.13 |
| ZipCache | 52.46 | 43.75 | 46.39 | 33.39 | 20.97 | 39.38 |
| ArkVale | 51.68 | 38.52 | 51.90 | 26.94 | 24.37 | 38.68 |
| **Sim-LLM(Ours)** | 42.66 | 28.87 | 48.68 | 45.38 | 21.57 | **37.43** |
| Llama-7B | 60.83 | 40.67 | 68.67 | 38.89 | 33.03 | 48.41 |
| H2O | 57.46 | 43.75 | 52.39 | 35.39 | 21.97 | 40.05 |
| StreamingLLM | 58.22 | 41.11 | 50.24 | 36.67 | 22.02 | 41.65 |
| ZipCache | 60.83 | 40.67 | 68.67 | 38.89 | 33.03 | 46.55 |
| ArkVale | 57.74 | 51.00 | 60.52 | 33.12 | 31.15 | 45.58 |
| **Sim-LLM(Ours)** | 58.45 | 39.61 | 67.78 | 36.19 | 32.43 | **46.89** |
| Llama-13B | 62.50 | 60.46 | 75.87 | 46.76 | 49.47 | 58.61 |
| H2O | 60.73 | 54.41 | 71.86 | 36.18 | 44.73 | 52.89 |
| StreamingLLM | 61.35 | 47.38 | 73.66 | 46.03 | 40.51 | 51.97 |
| ZipCache | 61.31 | 57.33 | 74.56 | 46.16 | 47.77 | 56.32 |
| ArkVale | 63.28 | 50.15 | 76.18 | 45.23 | 52.55 | 56.67 |
| **Sim-LLM(Ours)** | 61.47 | 58.74 | 73.66 | 45.89 | 46.81 | **57.31** |
| InternLM2-7B | 61.89 | 71.18 | 76.57 | 64.33 | 69.58 | 68.71 |
| **Sim-LLM(Ours)** | 60.19 | 69.22 | 74.47 | 61.01 | 65.12 | **66.00** |

Table 3: Maximum achievable batch size and throughput across different sequence lengths on NVIDIA A40 (40GB) servers and an A100 (80GB) GPU server. Notation "u + v" represents a prompt length of u and a generation length of v.

| GPU | Model Size | Seq. Length | Batch Size | | | | Throughput (tokens/s) | | | |
|---|---|---|---|---|---|---|---|---|---|---|
| | | | Llama | ZipCache | ArkVale | **Sim-LLM(Ours)** | Llama | ZipCache | ArkVale | **Sim-LLM(Ours)** |
| A40 | 1.1B | 5+8187 | 48 | 112 (2.3×) | 123 (2.6×) | 384 **(8×)** | 1424.96 | 2536.80 (1.8×) | 3584.00 (2.5×) | 4113.37 **(2.9×)** |
| | | 5+2043 | 239 | 448 (1.9×) | 768 (3.2×) | 1150 **(4.8×)** | 5142.86 | 8928.57 (1.7×) | 7428.57 (1.4×) | 10033.40 **(2.0×)** |
| | 7B | 5+128 | 128 | 224 (1.8×) | 243 (1.9×) | 640 **(5.0×)** | 568.56 | 1023.41 (1.8×) | 1228.09 (2.1×) | 1364.50 **(2.40×)** |
| | | 5+512 | 62 | 204 (3.3×) | 254 (4.1×) | 512 **(8.26×)** | 354.53 | 1985.37 (5.6×) | 2187.65 (6.2×) | 3816.51 **(10.76×)** |
| | | 5+2043 | 5 | 24 (4.8×) | 32 (6.4×) | 64 **(12.8×)** | 140.88 | 320.00 (2.27×) | 448.24 (3.18×) | 534.02 **(3.8×)** |
| | | 512+512 | 9 | 48 (5.3×) | 52 (5.7×) | 95 **(10.6×)** | 225.31 | 415.79 (1.8×) | 473.15 (2.1×) | 678.35 **(3.0×)** |
| | | 512+4096 | 7 | 96 (13.7×) | 108 (15.4×) | 256 **(36.6×)** | 65.25 | 246.30 (3.7×) | 277.16 (4.2×) | 522.53 **(8.0×)** |
| A100 | 7B | 2048+2048 | 15 | — | — | 128 **(8.5×)** | 141.10 | — | — | 421.02 **(3.0×)** |
| | 13B | 2048+2048 | 1 | — | — | 32 **(32×)** | 14.10 | — | — | 108.29 **(7.7×)** |

## B.4 Detailed Downstream Task Results

To further elucidate the findings presented in Figure 10b, we provide comprehensive statistical evaluations of downstream task performance in Table 4.

Table 4: Detailed downstream task results on Llama-7B

| # of Source Layer | HellaSwag | PIQA | BoolQ | Average |
|---|---|---|---|---|
| 3 | 41.76 | 67.9 | 57.28 | 55.65 |
| 5 | 42.14 | 66.97 | 61.47 | 56.86 |
| 14 | 43.43 | 68.17 | 59.57 | 57.06 |
| 16 | 43.88 | 67.57 | 61.07 | 57.50 |
| 30 | 44.22 | 68.28 | 60.73 | 57.74 |
| 32 | 44.74 | 69.21 | 61.88 | 58.61 |
| Standard model | 46.58 | 68.93 | 61.46 | 58.99 |

**Algorithm 1:** Sim-LLM

---

**Input** : Task batch $\mathcal{B} = \{t_1, t_2, \ldots, t_n\}$; cosine similarity threshold $\theta$; LSH bucket size $k$;
KV_Manager with cache size $C$ and LRU eviction; bottom stage num $sandwich\_bot$;
top stage num $sandwich\_top$.

**Output** : Inference results with KV optimization.

  ; // Step 1:  Preprocessing (§3.2)

**1 foreach** $t \in \mathcal{B}$ **do**
**2**     tokenize $t$ and compute embedding $e_t$;
**3**     $h_t \leftarrow \text{LSH}(e_t, k)$;
**4**     assign $t$ to bucket LSH_Bucket$[h_t]$;

  ; // Step 2:  Task Similarity Identification (§3.2)

**5 foreach** $t \in \mathcal{B}$ **do**
**6**     $S \leftarrow$ LSH_Bucket$[h_t]$;
**7**     compute $\text{sim}(e_t, e_s)$ for all $s \in S$;
**8**     **if** $\exists s^\star \in S$ *s.t.* $\text{sim}(e_t, e_{s^\star}) \geq \theta$ **then**
**9**       retrieve cached top-layer KV from KV_Manager$[s^\star]$;
**10**      $t.\text{KV} \leftarrow$ KV_Manager$[s^\star]$;
**11**     **else**
**12**       compute full KV for $t$ from scratch;

  ; // Step 3:  KV Cache Management (§3.3)

**13 foreach** *processed task* $t$ **do**
**14**     store $(e_t, h_t, \text{KV}_t)$ into KV_Manager;
**15**     **if** $\text{size}(KV\_Manager) > C$ **then**
**16**       apply LRU eviction;

  ; // Step 4:  Edge Server Communication (§3.4)

**17 foreach** $t \in \mathcal{B}$ *with no local match* **do**
**18**     compute task feature $f_t$ and compare with prototypes in global feature table;
**19**     **if** *remote feature match $f$ found* **then**
**20**       offload $t$ to the corresponding server;

**21** periodically update and sync task prototypes across servers;

  ; // Step 5:  Inference Execution (§3.3&3.4)

**22 foreach** $t \in \mathcal{B}$ **do**
**23**     **if** $t.KV$ *exists* **then**
**24**       run inference by reusing top-layer KV;
**25**     **else**
**26**       run inference with sandwich configuration, retaining bottom $sandwich\_bot$ layers and
        top $sandwich\_top$ layers KV;

---

## C  Algorithm Details

We provide the algorithm details of Sim-LLM in Algorithm 1. For each batch of tasks, after preprocessing, tasks are mapped to an LSH bucket, where similarity matching is performed. If a match is found, the KV of the similar task is reused to accelerate inference; otherwise, normal inference is performed with sandwich configuration. The hash value, embedding value and top-layer KV for each processed task are stored in the KV_Manager for future task reuse (LRU eviction is adopted when reaching cache size). After each batch is processed, each server updates its task prototype and sends it to other servers to maintain the global feature table.

