# OpenReview forum: "Sim-LLM: Optimizing LLM Inference at the Edge through Inter-Task KV Reuse"
_NeurIPS.cc/2025/Conference — NeurIPS 2025 poster_

### Official Review · Reviewer_vsRh · 2025-06-17

**Clarity:** 4
**Significance:** 4
**Originality:** 3
**Rating:** 5
**Confidence:** 4

**Summary:**

Sim-LLM is a novel framework exploiting the similarities across LLM inference requests to optimize the reuse of KV-cache.
The paper first identifies the similarity in requests across different inference tasks.
Such similarity allows the possibility of reusing previously generated KV-Cache.
The work also identifies the tradeoff between generation quality v.s reuse opportunity.

**Questions:**

This is a great work and I can also see great potential of this work in large-scale LLM infrastructure.

1. While this work targets edge devices, large LLM services also face a similar problem (each LLM customer often issues many requests on similar topics). Is there anything that needs special handling if we want to apply SimLLM to large LLM clouds?

2. How will the performance-accuracy tradeoff in Figure 11 look like in larger models? I believe this work should not be limited to small models and edge devices.

3. Is it possible to further optimize the performance by combining SimLLM with works that exploit the KV-cache sparsity (e.g., the baseline designs you compared with), so that SimLLM reuses the KV cache and maintains only the KV-cache of important tokens. This should generate a comprehensive tradeoff space between throughput performance and accuracy, where users can pick different tradeoff points in different scenarios.

**Ethical Concerns:**

["NO or VERY MINOR ethics concerns only"]

**Final Justification:**

All concerns well solved in the rebuttal.
The data presented in this paper (including appendix) and the rebuttal can well justify the motivation and effectiveness of the idea.
The presentation can still be somewhat improved (see final suggestion).
Overall, I am very positive about the paper.

I think it is totally fine to go for a simple idea instead of a complex one when possible, especially when this simple idea is useful and educative.

**Limitations:**

See questions, can this work apply to 1) large-scale LLM infras, 2) larger LLMs, or 3) other lossy KV-cache optimization techniques?

**Paper Formatting Concerns:**

no concern

**Quality:**

3

**Strengths And Weaknesses:**

Strength

While KV reuse is a well-explored field, this work extends the lossless reuse in previous works to the lossy reuse in this work. This enables a new tradeoff between generation quality and KV reuse opportunity.

Very clear motivation with quantified similarity across different tasks.

Significant performance benefits.


Weakness

The accuracy loss seems a little high if we want more performance benefits. So is there anyway to achieve better tradeoff between accuracy and performance?

---

> ### Author Rebuttal · Authors · 2025-07-31
>
> We deeply appreciate your recognition and careful review of our work — it truly means a lot to us. In response to your comments, we provide the following clarifications. Please let us know if any further clarification is needed.
>
> ---
>
>
>
> -   ## Response to Weakness
>
>  In our current design, we prioritized performance by optimizing **throughput** and **latency**, which can sometimes result in a slight **accuracy loss**. This trade-off was made in order to meet the strict **real-time** **processing** requirements of edge computing environments, where low latency and high throughput are often more critical than perfect accuracy.
>
> To address this trade-off, we are considering a **hybrid model architecture** that assigns high-accuracy models to critical tasks while using more efficient models for less critical ones. This would allow the system to maintain performance without sacrificing essential accuracy. Additionally, to further improve accuracy without compromising performance, we are exploring lightweight **post-processing techniques** to refine initial outputs, aiming to recover accuracy without significant performance degradation.
>
> ---
>
>
>
> -   ## Response to Q1
>
> Our motivation stems from the observation of task similarities commonly present in broad edge computing scenarios. In cloud environments, the larger scale of nodes and the greater diversity of tasks may pose challenges for identifying such task similarities. However, in specialized domains—such as medicine or law—where large language models are fine-tuned for domain-specific tasks, task similarity can still be high, making Sim-LLM fully applicable in these contexts. Moreover, the presence of high-speed data center networks in cloud infrastructures enables efficient inter-node communication. As a result, even without applying the communication optimizations proposed in Sim-LLM, the system still holds strong potential for performance gains when deployed in cloud settings.
>
> ---
>
>
>
> -   ## Response to Q2
>
>  Given our focus on edge scenarios, the primary model used in our experiments is Llama-7B. In response to your suggestion, we conducted **supplementary experiments with Llama-13B**, and the results are presented in the table below. As you anticipated, Sim-LLM maintains strong performance with the larger model, consistently outperforming smaller models such as Llama-7B and TinyLlama-1.1B. This observation aligns with the trends illustrated in **Figure 11 of Appendix B.1**. These findings further demonstrate the scalability and effectiveness of Sim-LLM when applied to larger language models.
>
> | Cos Threshold | Throughput(token/s) |  PPL  |  Acc  |
> | ------------- | :-----------------: | :---: | :---: |
> | 0.1           |       610.45        | 15.80 | 58.12 |
> | 0.2           |       608.32        | 15.72 | 58.45 |
> | 0.3           |       602.18        | 15.63 | 59.03 |
> | 0.4           |       595.67        | 15.38 | 60.21 |
> | 0.5           |       590.12        | 15.07 | 61.08 |
> | 0.6           |       585.44        | 14.54 | 61.32 |
> | 0.7           |       580.23        | 14.12 | 61.77 |
> | 0.8           |       575.89        | 13.81 | 62.54 |
> | 0.9           |       570.01        | 13.53 | 63.01 |
>
> ---
>
>
>
> -   ## Response to Q3
>
> As mentioned in *the third paragraph* of **Section 1** and in the discussion"*To address the second challenge*" in **Section 2**, **Sim-LLM is inherently compatible with intra-layer** **KV** **optimization** **techniques**. While existing methods primarily focus on KV optimization from a layer-wise or token-wise perspective, our work takes a task-wise approach. These perspectives are orthogonal, and therefore complementary. A promising direction for future work is to explore the integration of Sim-LLM with these fine-grained KV optimization strategies to further reduce memory consumption and mitigate accuracy degradation in edge environments.

---

> > ### Comment · Reviewer_vsRh · 2025-08-01
> >
> > Thanks for your response!
> >
> > I do like the simple but useful idea of leveraging request similarity to reduce computation cost, and want to see its potential in other LLM inference scenarios that have request similarity in your future work!
> >
> > However, for Weakness, if a significant loss of accuracy is an inevitable side product, maybe it is better to justify why your level of accuracy loss will not significantly impact the functionality of many edge LLM applications.

---

> > > ### Author Response · Authors · 2025-08-02
> > > **About Accuracy Loss**
> > >
> > > Thank you very much for your recognition and valuable suggestions. Your understanding is absolutely correct. Current mainstream KV inference acceleration techniques—such as eviction-based methods (e.g., H2O and StreamingLLM) and quantization-based methods (e.g., ZipCache)—inevitably involve a trade-off between performance and accuracy. Compared to the original models, these approaches generally cannot achieve significant speed improvements without sacrificing accuracy.
> > >
> > > However, **we assert that Sim-LLM does not exhibit "significant" accuracy loss in any specific domain**. Sim-LLM consistently maintains both overall and task-specific accuracy, ensuring reliable output quality, as demonstrated in Table 1 of Appendix B.5. Although Sim-LLM incurs slight accuracy loss relative to the original Transformer, **its accuracy still outperforms many existing SOTA optimization methods**. In particular, on LLaMA-7B and LLaMA-13B, Sim-LLM maintains high accuracy, with **accuracy loss on downstream tasks kept within 2.0%**. While its accuracy on TinyLLaMA-1.1B is slightly lower than that of ZipCache and ArkVale, it achieves up to **21.85% speedup and 21.88% memory reduction**. These improvements increase to **39.40% and 34.65%**, respectively, on LLaMA-7B. Moreover, on both LLaMA-7B and LLaMA-13B, Sim-LLM outperforms all four compared SOTA methods in terms of accuracy (see **Table 1 in Appendix B.5**).
> > >
> > > From a deployment perspective, we believe that compared to inference latency, **slight accuracy degradation typically has limited impact on most edge applications**. Prior studies have shown that even small increases in latency can significantly reduce user experience. For example, a 0.5s increase in latency can reduce Google traffic by 20%, and a 0.1s latency increase at Amazon may lead to a 1% drop in revenue [1, 2]. In edge environments—such as **smart assistants (e.g., Siri, smart speakers), in-vehicle voice interaction systems, and real-time monitoring and alerting systems**—research has also demonstrated that **even small increases in latency can significantly degrade user satisfaction** [3–5], whereas **a 2%–3% accuracy loss is often imperceptible to users** [6, 7]. Therefore, Sim-LLM is particularly well-suited for edge tasks that are delay- and resource-sensitive but tolerant to minor accuracy degradation.
> > >
> > > More importantly, **Sim-LLM supports flexible parameter control**, enabling dynamic trade-offs between performance and accuracy based on specific application requirements. For tasks requiring higher accuracy—such as **medical image analysis, legal document summarization, and financial question answering**—Sim-LLM can configure a **higher cosine similarity threshold** and **larger cache size** to minimize accuracy loss. This flexible design allows Sim-LLM to adapt to a wide range of edge applications and achieve **elastic inference** tailored to different scenarios.
> > >
> > > We sincerely appreciate your insightful comments. In the revised version, we will expand the discussion regarding the impact and controllability of accuracy loss. Please let us know if there is anything we can clarify further.
> > >
> > > **References**
> > >
> > > [1] QoS-Aware Content Delivery in 5G-Enabled Edge Computing: Learning-Based Approaches. TMC, 2024
> > >
> > > [2] OnRL: Improving mobile video telephony via online reinforcement learning. ACM MobiCom, 2020.
> > >
> > > [3] A provider-side view of Web search response time. ACM SIGCOMM, 2013.
> > >
> > > [4] UnFaaSener: Latency and cost aware offloading of functions from serverless platforms. USENIX ATC, 2023.
> > >
> > > [5] AIQoSer: Building the efficient inference-QoS for AI services. IEEE IWQoS, 2022.
> > >
> > > [6] QoE Assessment Model Based on Continuous Deep Learning for Video in Wireless Networks. IEEE TMC, 2023.
> > >
> > > [7] Profit Maximization of Delay-Sensitive, Differential Accuracy Inference Services in Mobile Edge Computing. IEEE TMC, 2025.

---

> > > > ### Comment · Reviewer_vsRh · 2025-08-02
> > > >
> > > > Thanks for your reply!
> > > >
> > > > Your answer has fully addressed all my concerns.
> > > >
> > > > A suggestion is you may consider presenting the information in Figure 11 & 12 in this way:
> > > > 1) You can draw a Pareto Optimal curve between throughput and accuracy using the data in fig.11 to show your tradeoff space.
> > > > 2) Plot existing work's throughput and accuracy as a point (if the point is under your curve, then it is easy to demonstrate why your work is better).
> > > > 3) Then, when you vary the similarity portion, you get different Pareto curves of simllm and different points of the existing work. This forms a 3d figure with axes on throughput, accuracy, and similarity. Your Pareto curves can form a 2D "Pareto plane", and the existing work's points can form a curve.
> > > > 4) Mark the similarity portion values of common edge workloads on the similarity axis. If we see that the existing work's curve is under your "Pareto plane" given common similarity values, then it is easy to visually show that your work is useful for common workloads.

---

> > > > > ### Author Response · Authors · 2025-08-02
> > > > >
> > > > > We sincerely thank you for this valuable suggestion. It offers a fresh perspective on how to present our results, and we truly appreciate your thoughtful input. Your feedback is highly encouraging and meaningful to us.

---

### Official Review · Reviewer_NQsy · 2025-06-30

**Clarity:** 3
**Significance:** 2
**Originality:** 2
**Rating:** 5
**Confidence:** 3

**Summary:**

This paper presents Sim-LLM, a novel approach for optimizing LLM inference on edge computing nodes by leveraging task similarity to reduce KV cache memory consumption. The key insight is that edge computing systems often process similar tasks due to geographical and temporal locality. The system caches KV pairs from previously processed tasks and reuses them for subsequent similar tasks, using cosine similarity and locality-sensitive hashing (LSH) for efficient similarity detection. The approach supports both single-node and multi-node scenarios with cross-server KV sharing.

**Questions:**

1. How does the approach scale when the number of cached tasks becomes very large? What are the computational and memory limits?
2. Can the system adapt to changing task distributions over time, or does it require periodic retraining of similarity models?
3. Are there any security implications of sharing KV caches across servers, particularly in edge computing contexts?
4. Does temperature sampling—where later tokens drift semantically—break the assumption that early-task KV is reusable?
5. What is the memory footprint of the global prototype table over a long time of edge traffic, and how is it pruned?

**Ethical Concerns:**

["NO or VERY MINOR ethics concerns only"]

**Final Justification:**

The authors clearly answered all my questions in the rebuttal. Their responses helped clarify the method and confirmed the validity of their results. I now believe the paper is solid and worth accepting.

**Limitations:**

1. Only consider a coarse granularity of the presentations among layers; layer importance also depends on the input prompt.
2. All servers must host identical weights because KV shapes must match.
3. If an edge node holding popular KV crashes, the benefit disappears until caches warm up again.
4. Performance is highly dependent on actual task similarity patterns, which may vary unpredictably.
5. The title is for optimizing LLM inference on the edge, but no tests for a single edge device.

**Paper Formatting Concerns:**

No concerns

**Quality:**

3

**Strengths And Weaknesses:**

Strengths
1. The observation on the edge computing systems is well-founded and empirically validated across multiple datasets.
2. The problem that this paper addressed is important.
3. It is the first work that reduces the KV cache memory consumption by considering the task similarities.
4. Extensive experiments to support the idea.

Weaknesses
1. The absence of experiments on genuine edge hardware limits the study’s relevance to practical edge-computing deployments.
2. The core insight about task similarity in edge computing is straightforward.
3. Algorithmic details (hash size, bucket eviction, sandwich cache policy) are scattered in text instead of one clear algorithm box.
4. For the multi-node setting, prototypes are exchanged frequently. The paper does not specify eviction, aging, or memory-bounded maintenance policies, raising questions about scalability over long-running edge workloads.

---

> ### Author Rebuttal · Authors · 2025-07-31
>
> We would like to express our sincere gratitude to the reviewer for your time and valuable feedback. We are especially thankful for your recognition of our work as “the first” in this direction and for highlighting our “well-founded observation.” Your recognition of the novelty of our work is aligns with the positive assessments provided by other reviewers. Below, we provide clarifications in response to your concerns. Please let us know if any further explanation is needed. **("W" refers to "Weakness" , "Q" refers to "Questions")**
>
> ---
>
>
>
> -   ## Response to W1
>
>   In practice, edge servers are typically deployed near users. While not as powerful as cloud servers, enterprise-grade edge deployments often use high-performance hardware. For instance, the NVIDIA Jetson AGX Orin offers 275 TOPS with a 12-core ARM CPU and 64GB DDR5 memory, and the Huawei Atlas 500 Pro provides up to 420 TOPS with a 24-core KunPeng 920 CPU, 128GB DDR4 memory, and 4 Atlas 300I GPUs. In our experiments, each edge server is configured with 4 NVIDIA A40 GPUs (299 TOPS), closely mirroring real-world setups and thus ensuring that our results reflect practical edge computing capabilities.
>
> ---
>
>
>
> -   ## Response to W2
>
> Although it may seem intuitive that similar tasks share similar KVs, **this insight has not been experimentally validated before**. In edge environments, task similarity for inference is often unknown, and we are **the first to demonstrate it experimentally**. Moreover, while most related works optimize inference at the token or layer level, our approach focuses on task-wise KV reuse, distinguishing it from conventional methods. This task-wise reuse effectively extends prior concepts of lossless reuse to lossy reuse, as noted by **Reviewer G55H and vsRh**.
>
> ---
>
>
>
> -   ## Response to W3
>
> To address your concerns, we now provide the **algorithm box** for Sim-LLM. For each task batch, tasks are mapped to LSH buckets for similarity matching. If a match is found, the corresponding KV is reused to accelerate inference; otherwise, standard inference with the sandwich configuration is applied. The hash, embedding, and top-layer KV are stored in the **KV_Manager** with **LRU** **eviction policy**. After each batch, servers update and exchange task prototypes to maintain a global feature table.
>
> ---
> **Input**:
> - Task batch `B = {t1, t2, ..., tn}`
> - Cosine similarity threshold `θ`
> - LSH bucket size `k`
> - KV_Manager (with cache size `C`, eviction policy: `LRU`)
> - bottom stage num `sandwich_bot`, top stage num `sandwich_top`
>
> **Output**:
>
> - Inference results with KV optimization
> ---
> **Procedure**:
> 1. **Preprocessing** (**Section 3.2**)
>
>     For each task `t∈B`:
>     - Tokenize input and generate embedding `e_t`
>         - Compute LSH hash `h_t ← LSH(e_t, k)`
>         - Assign `t` to bucket `LSH_Bucket[h_t]`
> 2. **Task Similarity Identification** (**Section 3.2**)
>
>     For each task `t∈B`:
>     - Search `LSH_Bucket[h_t]` for candidate tasks `S`
>         - For each candidate `s∈S`:
>         - Compute cosine similarity `sim(e_t, e_s)`
>         - If `∃s∈S` such that `sim(e_t, e_s) ≥ θ`:
>             - Retrieve cached top-layer KV from `KV_Manager[s]`
>                 - Assign `t.KV ← KV_Manager[s]`
>         - Else:
>             - Compute full KV for `t` from scratch
> 3. **KV Cache Management** (**Section 3.3**)
>     For each processed task `t`:
>     - Store `(e_t, h_t, KV_t)` in `KV_Manager`
>         - If cache exceeds size `C`, apply eviction policy (LRU)
> 4. **Edge Server Communication** (**Section 3.4**)
>
>     For task `t` with no local match:
>     - Compute task feature `f_t` and compare with prototypes in global feature table
>         - If remote feature match `f` found:
>             - Offload `t` to corresponding server
>     - Periodically update and sync task prototypes across servers
> 5. **Inference Execution** (**Section 3.3&3.4**)
>     a. For each task `t∈B`:
>     - If `t.KV` exists:
>         - Run inference by reusing top-layer KV
>     - Else:
>         - Run inference with `sandwich`configuration, retaining bottom `sandwich_bot` layers and top `sandwich_top` layers KV.
> ---
> -   ## Response to W4
>
> Each server maintains a global feature table containing task prototypes from all edge servers. After each batch, servers update their own prototypes and share them to keep global tables synchronized. These prototypes serve as lightweight indexes for efficient cross-server task matching. The global table is only updated upon receiving new prototypes and does not involve eviction.
>
> In contrast, the **KV_Manager** stores metadata for newly processed tasks and triggers **LRU** **eviction** when the cache limit is reached. We conduct new experiment to confirm the robustness of Sim-LLM. The detailed results can be found in our **response to Q3 for Review aP8N**.
>
> ---
>
>
>
> -   ## Response to Q1
>
> We conducted supplementary experiments to evaluate the latency and memory usage of Sim-LLM under a large cache size. The results are presented in **Table 1**.
>
> As the number of cached tasks increases, memory consumption grows accordingly. To manage this, we implement an **LRU** **eviction policy**, which effectively removes less frequently used entries, avoiding memory overflow and maintaining cache relevance.
>
> Although a larger cache may introduce overhead in task matching and KV lookups, we mitigate this using **locality-sensitive hashing (LSH)** for efficient approximate matching. Additionally, the **global feature table** accelerates semantic similarity matching across edge servers, keeping overall latency low. For instance, even with a cache size of 4096, Sim-LLM achieves a latency of **45.35ms**, compared to **60.61ms** for LLaMA.
>
> |Cache Size|Latency (ms)| Memory (GB) |
> |---|---|---|
> |2048|33.13|14.98|
> |2560|36.77|16.91|
> |3584|41.97|22.87|
> |4096|45.35|25.50|
>
> **Table 1** Latency and memory usage under large cache sizes.
>
> ---
>
>
>
> -   ## Response to Q2
>
> (1)  We conducted new experiments to evaluate system performance under different request distributions—Poisson (uniform) and Power-law (bursty)—focusing on both perplexity and throughput. As shown in **Table 2**, Sim-LLM maintains high performance across varying task patterns. Under bursty workloads, similar tasks occur more frequently, enabling the LRU-based KV_Manager to retain and reuse KVs more effectively, thereby accelerating inference. These results demonstrate that Sim-LLM adapts well to dynamic task distributions.
>
> |Cache Size|Throughput (token/s) (Uniform \| Bursty)| PPL (Uniform\| Bursty) |
> |---|---|---|
> |256|545.03\|435.78|11.19\|11.21|
> |512|549.58\|434.03|10.76\|10.89|
> |1024|547.92\|428.09|10.49\|10.61|
> |2048|517.92\|426.63|10.31\|10.44|
>
> **Table 2** Performance under different task distributions.
>
> (2) **Our method does not require periodic retraining**, as Sim-LLM relies on cosine similarity combined with LSH for semantic matching, rather than a trainable similarity model. This design significantly accelerates inference without requiring costly architectural modifications and allows the system to dynamically adapt to new tasks, ensuring both efficiency and scalability.
>
> ---
>
>
>
> -   ## Response to Q3
>
>  Similar to many prior works such as [1] [2] [3], our approach assumes that all edge nodes are trustworthy for KV cache sharing. Nonetheless, we acknowledge the security challenges inherent in cross-server communication within edge computing environments. Malicious nodes could threaten data integrity and confidentiality. Existing research addresses these concerns, such as [4], which analyzes potential attacks and defense strategies at the edge; and [5], which enforces secure data sharing via access control mechanisms.
>
> While our work does not focus on securing cross-node communication, existing security solutions can be integrated to mitigate these risks. We consider security issues orthogonal to our core contributions—task acceleration and KV reuse—and thus not directly impacting the proposed methods.
>
> ---
>
>
>
> -   ## Response to Q4
>
>  Temperature sampling controls output randomness by scaling logits before selecting the next token. Higher temperatures increase randomness, while lower temperatures yield more deterministic outputs.
>
> Adjusting the temperature impacts only the quality of the tokens generated during the current step and does not alter previously generated tokens. Consequently, if the existing tokens match a stored similar task, the associated KV cache can still be reused. Therefore, the reusability of early-task KVs remains unaffected.
>
> ---
>
>
>
> -   ## Response to Q5
>
> The global prototype table stores task prototypes that serve as local indexes for efficient cross-server task matching. Its memory usage depends only on the number of edge servers. To address your concern, we conducted supplementary experiments (**Table 3**) measuring memory consumption over time. Due to the inherent similarity of task features and fluctuating traffic in edge environments, prototype updates are minimal, and memory usage stabilizes.
>
> Each edge server maintains one such table. After processing each batch, a edge server updates its prototype and shares the update with other edge servers to keep their global prototype tables synchronized. Only when an edge server leaves the network, the item corresponding to the edge server will be pruned.
>
> |Time Slot|Memory Usage (KB)|
> |---|---|
> |100th|1688|
> |200th|1727|
> |300th|1692|
> |400th|1709|
>
> **Table 3** Memory usage over time.
>
> ---
>
>
>
> ## Refs
>
> [1] OL-MEDC: An Online Approach for Cost-Effective Data Caching in Mobile Edge Computing Systems. TMC, 2023.
>
> [2] Joint service placement and request routing in multi-cell mobile edge computing networks. INFOCOM, 2019.
>
> [3] Service routing in multi-tier edge computing: A matching game approach. JSAC, 2022.
>
> [4] Edge Computing Security: State of the Art and Challenges. Proceedings of the IEEE, 2019.
>
> [5] Everything Under Control: Secure Data Sharing Mechanism for Cloud-Edge Computing. TIFS, 2023.

---

### Official Review · Reviewer_G55H · 2025-07-01

**Clarity:** 3
**Significance:** 3
**Originality:** 3
**Rating:** 4
**Confidence:** 2

**Summary:**

The paper introduces Sim-LLM, a novel method to optimize Large Language Model (LLM) inference in edge computing systems by reusing Key-Value (KV) caches from similar tasks. Sim-LLM caches top-layer KVs of processed tasks and reuses them for subsequent similar tasks, reducing redundant computations. The system employs Locality-Sensitive Hashing (LSH) for efficient task similarity identification and a KV Manager to handle cache storage and sharing across edge servers.

**Questions:**

See weaknesses.

**Ethical Concerns:**

["NO or VERY MINOR ethics concerns only"]

**Final Justification:**

The authors provide detailed clarifications about the inter-server communication and the similarity threshold. I have no further questions and will keep my rating.

**Limitations:**

See weaknesses.

**Quality:**

3

**Strengths And Weaknesses:**

Strengths:
1) The paper reduces KV cache memory overhead by introducing task similarity-aware reuse, which is a direction underexplored in prior work.
2) The paper is well-motivated and the observations are empirically compelling.
3) The method demonstrates up to 34.65% memory reduction and 39.40% throughput improvement on A40/A100 GPUs.

Weaknesses:
1) The multi-node KV sharing mechanism assumes low-latency inter-server communication, which may not hold in real-world edge networks.
2) While the cosine similarity threshold is justified empirically, it lacks a theoretical or intuitive explanation for why this value generalizes across diverse edge workloads.

---

> ### Author Rebuttal · Authors · 2025-07-31
>
> We would like to express our sincere gratitude to the reviewer for your time and the valuable feedback. In particular, your recognition of our work as “a novel method”，“a direction underexplored in prior work” and “well-motivated” aligns with the positive assessments provided by other reviewers. Below, we provide clarifications in response to your concerns. Please let us know if any further explanation is needed. **("W" refers to "Weakness" , "Q" refers to "Questions")**
>
> ---
>
>
>
> -   ## **Response to W1**
>
> We would like to highlight that, in practice, many **edge computing environments** are already leveraging **high-speed communication links** between servers [1] [2] [3] [4] [5], including InfiniBand which is a high performance technology aiming to provide low-latency and high-bandwith. In our paper, the edge servers are connected via an InfiniBand network to facilitate the exchange of task prototypes and KV sharing.
>
> To minimize communication overhead due to information exchange between edge servers, when similar tasks are identified on another edge server, only the sequence or the corresponding word embeddings are transmitted to the identified edge server, rather than transmitting the KV cache of the similar task to the current processing task from the identified server back to the original server. Furthermore, regarding the inter-server communication concerns you raised, we conducted supplementary quantitative experiments, as shown in **Table 1**. These results demonstrate that the communication overhead is almost negligible compared to the inference latency.
>
> | Node Num | Communication Latency (ms) | Inference Latency (ms) |
> | -------- | :------------------------: | :--------------------: |
> | 2        |            1.22            |         483.15         |
> | 4        |            2.73            |         337.84         |
> | 6        |            4.12            |         303.51         |
> | 8        |            8.71            |         272.19         |
> | 12       |           12.92            |         232.76         |
>
> **Table 1** Communication latency and inference latency with varying numbers of nodes.
>
> ---
>
>
>
> -   ## **Response to W2**
>
> In **Appendix B.1 Impact of** **Cosine Similarity** **Threshold**, we present experimental results illustrating how throughput, perplexity (ppl), and downstream task accuracy vary with different cosine similarity threshold. When the threshold is low, throughput performance exceeds that of a higher threshold. This is because most incoming tasks can easily identify "similar" tasks' KVs for reuse, introducing minimal identification overhead and accelerating the inference process. However, this gain comes at the cost of model accuracy. When the threshold is incresd to 0.9, there is a significant drop in generation speed due to fewer tasks being able to reuse KVs, resulting in many tasks performing inference as in the standard Transformer model. Therefore, as a trade-off between generation speed and model accuracy, a threshold of 0.8 is preferred.
>
> The generalizability of this threshold is supported by the observation that many edge tasks share underlying semantic similarities. As noted in **Observation 2 (Section 1)**, KV caches generated by similar tasks are themselves similar, indicating that task features exhibit consistent patterns across different edge scenarios. The selected threshold captures these shared characteristics, making it robust across varied workloads. Moreover, it is resilient to small fluctuations in task features, which are common in dynamic edge environments.
>
> To further validate the robustness of Sim-LLM, we conducted supplementary experiments using different thresholds (0.6, 0.7, 0.8, and 0.9) across four edge servers. The results, shown in **Table 2**, align with the trends observed in **Appendix B.4 Table 2**, demonstrating that Sim-LLM maintains consistent behavior under variable threshold settings. As expected, lower thresholds enable more KV reuse and faster inference, while higher thresholds reduce reuse and slow down inference. Nevertheless, due to the semantic consistency of tasks across servers, performance remains stable over a range of threshold values, confirming the adaptability of Sim-LLM in heterogeneous edge environments.
>
> | Seq. Length | Llama Throughput (token/s) | Different Threshold Throughput (token/s) | Uniform Threshold Throughput (token/s) |
> | :---------- | :------------------------: | :--------------------------------------: | :------------------------------------: |
> | 5+128       |           568.56           |           1280.78 **(2.25x)**            |          1364.50 **(2.40x)**           |
> | 5+512       |           354.53           |           3658.23 **(10.31x)**           |          3816.51 **(10.76x)**          |
> | 5+2043      |           140.88           |            422.88 **(3.00x)**            |           534.02 **(3.8x)**            |
> | 512+512     |           225.31           |            601.28 **(2.67x)**            |           678.35 **(3.0x)**            |
> | 512+4096    |           65.25            |            477.11 **(7.31x)**            |           522.53 **(8.0x)**            |
>
> **Table 2** Throughput comparison for different sequence lengths and thresholds.
>
> ---
>
>
>
> ## Refs
>
> [1] Learning-Aided Computation Offloading for Trusted Collaborative Mobile Edge Computing, IEEE TMC, 2020.
>
> [2] FlowStar: Fast Convergence Per-Flow State Accurate Congestion Control for InfiniBand. IEEE/ACM ToN, 2024.
>
> [3] Tutti: coupling 5G RAN and mobile edge computing for latency-critical video analytics. MobiCom, 2022.
>
> [4] C2DN: How to Harness Erasure Codes at the Edge for Efficient Content Delivery. NSDI, 2022.
>
> [5] Argus: Enabling Cross-Camera Collaboration for Video Analytics on Distributed Smart Cameras. IEEE TMC, 2025.

---

> > ### Comment · Reviewer_G55H · 2025-08-05
> > **Response to the rebuttal**
> >
> > Thank you for the detailed clarifications about the inter-server communication and similarity threshold. As I am not an expert in this field, I have no further questions and will keep my rating.

---

> > > ### Author Response · Authors · 2025-08-06
> > >
> > > Thank you for your positive response. We’re glad the clarifications and additional results addressed your concerns. Your feedback has been valuable in improving the quality of our paper. Thanks again!

---

> ### Author Response · Authors · 2025-08-05
>
> Dear Reviewer G55H
>
> As we near the end of the author-reviewer discussion phase, we would like to sincerely thank you again for your time and valuable feedback. If you think our clarifications have addressed your concerns, we would deeply appreciate your support in updating the score accordingly.
>
> If there are any remaining questions or points you would like us to clarify, please feel free to let us know. We’re here to support the discussion as best we can.
>
> Best regards,
>
> The Authors

---

### Official Review · Reviewer_aP8N · 2025-07-12

**Clarity:** 3
**Significance:** 2
**Originality:** 3
**Rating:** 3
**Confidence:** 4

**Summary:**

This paper identifies two novel observations: task similarity and KV cache similarity among LLM tasks at the edge. The authors identify similar tasks using cosine similarity, Locality-Sensitive Hashing (LSH) or prototype learning within and across edge servers, and reuses the top-layer KV cache from previous tasks to accelerate the inference.

**Questions:**

- When fewer tasks in a batch have similar cached tasks, what is the overhead of postponing them? Have you evaluated the worst-case latency in such situations?
- You choose 0.8 as the cosine similarity threshold. How sensitive are the results to this threshold in real-time inference scenarios? For instance, how does throughput or perplexity change with slightly higher or lower thresholds under typical workloads?
- What eviction policy is used for managing the KV cache (e.g., FIFO, LRU)? How does this policy affect performance under bursty or highly variable load, where task arrival patterns may not be uniform?
- In Figure 6, what type of network is used across nodes? Have you measured the communication overhead introduced by task prototype exchange and KV sharing? Could this overhead become a bottleneck in large-scale deployments?
- ⠀[Typo] Line 148: “such as MQA and GQA used in flash-attention” is inaccurate. MQA and GQA are architectural optimizations, while FlashAttention is dataflow optimization for the attention module. FlashAttention dataflow can be applied to models using MHA, MLA, MQA and GQA, but it is incorrect to say that MQA and GQA are “used” in FlashAttention.

**Ethical Concerns:**

["NO or VERY MINOR ethics concerns only"]

**Limitations:**

Yes

**Quality:**

3

**Strengths And Weaknesses:**

**Strength**
* This paper proposes solid observations and corresponding challenges about task and KV similarity, and supports them with concrete visualizations and quantitative results.
* The authors clearly illustrate their methods to address the challenges: using adaptive LSH for similarity detection, partial KV caching for unmatched tasks, and prototype-based routing for cross-node reuse.
* The method targets edge deployment scenarios with constrained memory, a highly relevant use case for real-world LLM serving.

**Weakness**
* While the overall idea of leveraging task similarity for KV cache reuse is novel, the core methodology is relatively simple and straightforward. The mechanism relies on matching semantic similarity via cosine similarity or LSH and reusing top-layer KV, which lacks algorithmic innovation and theoretical depth.
* For Challenge 2, the paper proposes skipping the KV computation of intermediate layers by caching only the top and bottom layer KV. However, the justification for this caching scheme is insufficient. The authors assume that skipping the middle layers preserves task semantics and output quality, but there is a lack of detailed theoretical analysis to support the validity.
* The empirical evaluation lacks fine-grained error analysis. There is no quantitative analysis of the success rate of KV reuse, i.e., how often tasks are matched and accelerated in real scenarios. While average accuracy and PPL are reported, the paper does not discuss how incorrect KV reuse (e.g., misclassified similar tasks) affects output quality, which could be critical in edge deployment scenarios.

---

> ### Author Rebuttal · Authors · 2025-07-31
>
> We would like to express our sincere gratitude to the reviewer for your time and valuable feedback. In particular, your recognition of our “solid observations” and the comment that “the overall idea is novel” align with the positive assessments provided by other reviewers. Below, we provide clarifications in response to your concerns. Please let us know if any further explanation is needed. **("W" refers to "Weakness" , "Q" refers to "Questions")**
>
>  ---
>
> -   ##  **Response to W1**
>
>
> We acknowledge that both cosine similarity and Locality Sensitive Hashing (LSH) are well-established techniques. Their combination, however, offers a scalable and efficient solution for task matching based on semantic similarity, significantly accelerating model inference without requiring costly architectural modifications. Cosine similarity, in particular, has a strong theoretical foundation and has been extensively studied [1] [2]. As discussed in [2], it can be interpreted as the sum of semantic similarities along embedding dimensions after Independent Component Analysis, further justifying its use for semantic comparison.
>
> We adopt top-layer KV reuse is motivated by the interpretation of the Transformer’s stacked architecture as an iterative refinement process, where lower layers tend to encode syntactic information, while higher layers focus more on semantics [3] [4]. In **Section 4.3**, we further validate this choice through experiments that compare the performance of KV reuse from different layers, demonstrating the superiority of using top-layer KV as the source.
>
>  ---
>
> -   ## **Response to W2**
>
>
>  As shown in [5] [6] [7], LLMs exhibit significant layer redundancy—intermediate layers contribute minimally and can be removed with limited performance degradation. This supports our design choice: even when skipping these layers, retaining caches from the bottom and top layers preserves essential task-relevant information. We further validate this through experiments in **Section 4.3**.
>
>
> ---
>
>
> -   ## **Response to W3**
>
>
>  We conducted additional experiments to examine how the KV reuse match rate changes with the number of tasks. As shown in **Table 1**, the match rate increases as the task set grows. The values in parentheses indicate the proportion of similar tasks in each set. These results suggest that a larger task pool provides more opportunities for KV reuse, thereby enhancing inference efficiency.
>
> |Number of Tasks|Match Rate (0.35)|Match Rate (0.65)|Match Rate (0.89)|
> |---|---|---|---|
> |500 |0.063|0.147|0.258|
> |1000|0.089|0.224|0.432|
> |1500|0.133|0.278|0.583|
> |2000|0.158|0.383|0.664|
> |2500|0.181|0.415|0.722|
>
> **Table 1** KV reuse match rate based on the number of tasks and varying similarity proportion of task sets.
>
>
>
> Regarding incorrect KV reuse, this situation occurs when LSH distortion happens, i.e., two dissimilar tasks are erroneously mapped to the same bucket, or similar tasks are mapped to different buckets. We first explored the probability of this occurrence, and the results, shown in **Table 2**, indicate that **this probability does not exceed 3%**.
>
> |Number of Tasks|Misclassified Task Num|Error Rate|
> |---|---|---|
> |1000|7|0.7%|
> |2000|15|0.75%|
> |3000|18|0.6%|
> |4000|37|0.925%|
> |5000|52|1.04%|
>
> **Table 2** Probability of incorrect KV reuse due to LSH distortion.
>
>
>
> Next, we simulated the impact of a higher error rate on the output quality by forcing tasks to map to the same LSH bucket. The experiments show that only when this probability exceeds 10% does it significantly affect the quality of the output. However, this probability is unlikely to occur in real-world scenarios, indicating that LSH distortion does not overly impact the model accuracy of Sim-LLM.
>
> |Error Rate|PPL|
> |---|---|
> |0.5%  |15.13|
> |0.75% |15.13|
> |1.00% |15.15|
> |2.00% |15.14|
> |3.00% |15.21|
> |5.00% |15.24|
> |7.00% |15.54|
> |10.00%|16.97|
>
> **Table 3** Impact of error rate on model output quality.
>
>
> ---
>
>
> -   ##  **Response to Q1**
>
>
> To improve overall inference efficiency, Sim-LLM postpones tasks that fail to match similar ones within a batch. This allows matched tasks to proceed with KV reuse without being delayed by unmatched tasks. If few tasks in a batch are eligible for reuse, the inference time remains comparable to that of standard processing, and no postponement occurs.
>
> To address the reviewer’s concern about the worst-case scenario—where unmatched tasks are repeatedly delayed across batches—we conducted an experiment to measure the probability of continuous postponement. As shown in **Table 4**, the probability of a task being postponed for 5 consecutive batches is only **1.1%**, indicating that such cases are **worst-case**. Additionally, **Table 5** reports the delay overhead introduced by postponement, showing minimal impact on overall inference efficiency (e.g., average inference time remains 0.094s for sequence length 5+128).
>
> |Postponed Times|Proportion|
> |---|---|
> |1|74.7%|
> |2|12.4%|
> |3|5.6%|
> |4|1.9%|
> |5|1.1%|
>
> **Table 4** Probability of tasks being continuously postponed across multiple batches.
>
>
>
> |Similarity Proportion of Task Sets|Latency by Postponing (s)|
> |---|---|
> |0.14|0.027|
> |0.35|0.058|
> |0.53|0.064|
> |0.65|0.077|
> |0.74|0.083|
> |0.89|0.109|
>
> **Table 5** Delay overhead introduced by postponed tasks with varying similarity proportions of task sets (sequence length is 5+128).
>
>
> ---
>
>
> - ## **Response to Q2**
>
>  As shown in **Appendix B.1 (Impact of** **Cosine Similarity** **Threshold)**, we analyzed how the similarity threshold affects throughput, perplexity, and downstream accuracy. A lower threshold improves throughput by enabling more tasks to reuse KVs with minimal overhead. However, this comes at the cost of accuracy. Conversely, at a high threshold (e.g., 0.9), KV reuse becomes rare, leading to slower inference comparable to standard Transformers. As a trade-off, we select 0.8 as the optimal threshold, balancing generation speed and model performance.
>
>  ---
>
> -   ## **Response to Q3**
>
>
>  (1) We adopt the LRU strategy to maximize reuse of similar KV pairs and thus accelerate inference. LRU retains frequently reused task KVs while evicting the least reused ones.
>
> (2) To address concerns about workload variability, we evaluated LRU under two task distributions: **Poisson (uniform workload)** and **Power-law (bursty workload)**. Results in **Table 6** show that under bursty conditions, LRU maintains strong performance without significant latency increase compared to uniform loads. This is because bursty workloads tend to have many tasks sharing similar features, allowing LRU to effectively retain frequently reused KVs and boost inference speed.
>
>
>
> |Cache Size|Poisson Latency (ms)|Power-law Latency (ms)|
> |---|---|---|
> |256|25.66|27.68|
> |512|25.75|28.51|
> |1024|26.31|29.34|
> |2048|33.13|35.88|
> |4096|45.35|47.72|
>
>
>
> **Table 6** Latency comparison for Poisson and Power-law distributions under the LRU strategy.
>
> ---
>
> -   ## **Response to Q4**
>
>
>  We use an InfiniBand network to connect edge servers for exchanging task prototypes and KV sharing, leveraging its low-latency, high-bandwidth characteristics widely adopted in edge computing [8] [9] [10].
>
> After processing each batch, edge servers update and exchange prototypes to maintain a global feature table. To reduce communication overhead, servers typically exchange task sequences or embeddings rather than full KV caches, as noted in the last paragraph of **Section 3.4**.
>
> Regarding the **communication overhead** introduced by task prototype exchange and KV sharing, we conducted new experiments. The results, as shown in **Table 7**, indicate that this overhead is negligible compared to the inference cost.
>
> |Batch Size|Communication Latency (ms)|Inference Latency (ms)|
> |---|---|---|
> |4|2.37|368.35|
> |8|2.43|310.52|
> |64|2.73|304.02|
> |128|2.88|349.59|
> |256|3.59|549.73|
> |1024|5.47|700.70|
> |2048|10.23|1311.18|
>
>
>
> **Table7** communication and inference latency for different batch sizes.
>
>
>
>  Additionally, we also conducted supplementary experiments to evaluate latency in **large-scale deployments**. The results presented in **Table 8** indict, although communication overhead tends to increase as the network scales up, its overall impact on inference efficiency remain limited.
>
> |Number of Nodes|Communication Latency (ms)|Inference Latency (ms)|
> |---|---|---|
> |2|1.22|483.15|
> |4|2.73|337.84|
> |6|4.12|303.51|
> |8|8.71|272.19|
> |12|12.92|232.76|
>
> **Table 8** Communication latency and inference latency with varying numbers of nodes.
>
>
> ---
>
>
> -   ## **Response to Q5**
>
>
>  Thank you for pointing out the typo. What we intend to convey is the potential integration between our work and approaches using MQA or GQA techniques. We will revise the statement accordingly in the manuscript.
>
>  ---
>
> ## **Refs**
>
> [1] Correlation Coefficients and Semantic Textual Similarity. NAACL, 2019.
>
>
>
> [2] Revisiting Cosine Similarity via Normalized ICA-transformed Embeddings. ACL, 2025.
>
>
>
> [3] Probabilistic Transformer: A Probabilistic Dependency Model for Contextual Word Representation. ACL, 2023.
>
>
>
> [4] BoolQ: Exploring the Surprising Difficulty of Natural Yes/No Questions. NAACL, 2019.
>
>
>
> [5] ShortGPT: Layers in Large Language Models are More Redundant Than You Expect. ACL, 2025.
>
>
>
> [6] LaCo: Large Language Model Pruning via Layer Collapse. EMNLP, 2024.
>
>
>
> [7] Llm-pruner: On the structural pruning of large language models. NIPS, 2023.
>
>
>
> [8] FlowStar: Fast Convergence Per-Flow State Accurate Congestion Control for InfiniBand. ToN, 2024.
>
>
>
> [9] Alarm: An Adaptive Routing Algorithm Based on One-Way Delay for Infiniband. TNSE, 2024.
>
>
>
> [10] Enabling the CUDA Unified Memory model in Edge, Cloud and HPC offloaded GPU kernels. CCGrid, 2022.
>
> [11] Tutti: coupling 5G RAN and mobile edge computing for latency-critical video analytics. MobiCom, 2022.

---

> ### Author Response · Authors · 2025-08-05
>
> Dear Reviewer aP8N
>
> As we near the end of the author-reviewer discussion phase, we would like to sincerely thank you again for your time and valuable feedback. If you think our clarifications have addressed your concerns, we would deeply appreciate your support in updating the score accordingly.
>
> If there are any remaining questions or points you would like us to clarify, please feel free to let us know. We’re here to support the discussion as best we can.
>
> Best regards,
>
> The Authors

---

> > ### Comment · Reviewer_aP8N · 2025-08-08
> >
> > Thank you for your response, which has addressed part of my concerns. However, I still believe that the acceleration approach of skipping computations in certain middle layers during inference is rather aggressive and lacks generalizability. Moreover, skipping intermediate layers amplifies the effect of reusing the KV cache from the top layers; without such skipping, the relative benefit of your KV reuse optimization would likely be smaller. In addition, deferring certain tasks within the same batch instead of processing them together may lead to underutilization of hardware resources. Therefore, I am inclined to maintain my current score.

---

> > > ### Author Response · Authors · 2025-08-08
> > > **Response to Further Comments**
> > >
> > > We sincerely appreciate your positive response, the valuable time you have dedicated, and the constructive suggestions that help improve our work. ***Regarding your further comments, we believe there might be some misunderstandings, and we would like to provide additional clarifications.***
> > > - ***KV Cache Reuse and Layer Skipping***
> > >
> > > The purpose of KV cache techniques is inherently to reduce redundant computation through reuse. The novelty of Sim-LLM lies in being the first to exploit **task-wise similarity in edge tasks and to provide in-depth experimental evidence showing its substantial potential for KV reuse**. This point has been consistently recognized by you and other reviewers—for example: **your remark** that our work “proposes solid observations,” **Reviewer G55H**’s comment on “a direction underexplored in prior work,” **Reviewer NQsy**’s note that the “observation is well-founded and empirically validated,” and **Reviewer vsRh**’s statement about “extending lossless reuse to lossy reuse.”
> > >
> > > Layer skipping is a one of the core designs of Sim-LLM: reusing top-layer KV of similar tasks effectively leverages the most informative semantic representation, thereby eliminating the need to compute intermediate layers whose contribution is relatively minor—a principle supported by prior works [1–4]. This mechanism and top-layer KV reuse are **mutually reinforcing**, rather than separate technical points. Therefore, the statement that “without such skipping, the relative benefit of your KV reuse optimization would likely be smaller” does not accurately reflect our design rationale.
> > >
> > > Furthermore, we have extensively validated this layer skipping strategy across **multiple models** (TinyLlama-1.1B, Llama-7B, Llama-13B, InternLM2-7B) and **diverse downstream tasks** (Reasoning, Language, Knowledge, Examination, Understanding), as shown in **Table 1 of Appendix B.3** of the manuscript. Sim-LLM does not impair model generalization; in fact, it demonstrates **better generalization** compared to state-of-the-art methods.
> > >
> > > - ***Postponing Tasks and Utilization of Hardware Resources***
> > >
> > > We appreciate your attention to our postponing mechanism. **We would like to clarify that this strategy does not reduce hardware utilization. On the contrary, it was designed to improve it.** For example, in a batch with a high proportion of similar tasks, if the few dissimilar tasks are not postponed to the next batch, KV reuse will cause similar tasks to finish early and wait for the dissimilar ones to complete, lowering overall resource utilization. Postponing these tasks also increases the likelihood of matching with similar-task KV in the next batch, thereby improving throughput (Figure 7(a), Section 4.2).
> > > As shown in **Table 2 of Appendix B.4**, Sim-LLM increases the **maximum achievable batch size**, which is in fact another key contribution. This demonstrates that Sim-LLM can handle more tasks in the same batch, further **improving hardware utilization**.
> > >
> > > *We apologize if our initial rebuttal caused any misunderstanding. Thank you again for your time and insightful feedback. Our goal is not merely to obtain a high score, but to refine our work through discussions such as this. We greatly appreciate your engagement and the opportunity to improve our contribution.*
> > >
> > >
> > > **Refs**
> > >
> > > [1] Probabilistic Transformer: A Probabilistic Dependency Model for Contextual Word Representation. ACL, 2023.
> > >
> > > [2] BoolQ: Exploring the Surprising Difficulty of Natural Yes/No Questions. NAACL, 2019.
> > >
> > > [3] ShortGPT: Layers in Large Language Models are More Redundant Than You Expect. ACL, 2025.
> > >
> > > [4] LaCo: Large Language Model Pruning via Layer Collapse. EMNLP, 2024.

---

### Comment · Area_Chair_qxTH · 2025-08-02

Dear Reviewers

The authors have responded to your reviews. In the next few days, please read their responses and engage in a productive discussion that will be critical to the review process.

I truly appreciate your timely thoughts and comments!

AC

---

### Author Response · Authors · 2025-08-09
**Response to all reviewers and AC**

We would like to express our sincere gratitude to the AC and all reviewers for their time, constructive feedback, and the overall positive evaluation of our work. We are especially encouraged by the recognition reflected in the following comments:

- **Reviewer aP8N**: "Solid observations," "clearly illustrates their methods to address the challenges," "the overall idea is novel."
- **Reviewer G55H**: "Well-motivated," "a direction underexplored in prior work," "observations are empirically compelling."
- **Reviewer NQsy**: "The observation is well-founded," "first work to reduce KV cache memory consumption by considering task similarities."
- **Reviewer vsRh**: "a great work and I can also see great potential of this work," "very clear motivation with quantified similarity across tasks," "significant performance benefits."

Inspired by the thoughtful comments from the reviewers, we have made the following clarifications to address their concerns:
1. Added experiments demonstrating that Sim-LLM incurs low communication overhead.
2. Provided a clearer explanation of the task postponing mechanism, supported by additional experiments and descriptions.
3. Included further experiments under both bursty and uniform task workloads.
4. Incorporated scalability experiments to evaluate performance in large-scale deployment scenarios.
5. Elaborated on the specific cache strategy and the overall algorithmic framework of Sim-LLM.

**As we approach the conclusion of the author–reviewer discussion, we would like to express our sincere gratitude to the AC and all reviewers for their active engagement and constructive feedback.** The reviewers’ thoughtful comments have guided us to refine our presentation, strengthen the theoretical underpinnings, and highlight the practical relevance of our work. We are also encouraged by the positive feedback received during the rebuttal period, which further motivates us to advance this line of research.

---

### Note · Authors · 2025-08-13

While most related works optimize inference at the token or layer level, their ability to reduce memory consumption remains limited when the number of tasks is large,  leading to significant computation overhead. **Sim-LLM is the first to exploit task-wise similarity in edge inference tasks**, leveraging such similarity to reduce KV cache memory consumption and boost inference efficiency. Building on **three key observations**, its **core idea** is to maximize the reuse of similar KVs to accelerate inference. In particular, Sim-LLM can **identify request similarities across nodes** in resource-limited edge computing systems.

We sincerely thank the AC and all reviewers for their time, constructive feedback, and positive evaluation of our work. We are particularly encouraged by the recognition of our approach as novel and well-motivated, with solid observations and empirical support **(Reviewer aP8N, G55H)**. Several reviewers highlighted our unique contribution in addressing task similarity to reduce KV cache memory consumption, a direction underexplored in prior research **(Reviewer NQsy, vsRh)**. Additionally, the potential of our work was acknowledged, with significant performance benefits noted across tasks **(Reviewer vsRh)**.

During the rebuttal phase, we strengthened Sim-LLM from both theoretical and empirical aspects:

- Added experiments demonstrating that Sim-LLM incurs low communication overhead.
- Incorporated additional clarification on the criteria for selecting the cosine similarity threshold and the reuse source layer.
- Clarified that Sim-LLM does not result in significant accuracy degradation.
- Provided a clearer explanation of the task postponing mechanism, supported by additional experimental descriptions. Clarified that Sim-LLM does not lead to lower hardware utilization; rather, it increases the maximum achievable batch size.
- Explained the mechanism of global feature table and its role in accelerating semantic similarity matching among edge servers to reduce overall computational costs.
- Included further experiments under both bursty and uniform task workloads.
- Conducted scalability experiments to evaluate performance in large-scale deployment scenarios.
- Elaborated on the specific cache strategy and the overall algorithmic framework of Sim-LLM

We believe that these clarifications significantly enhance the practical relevance of our work. **Once again, we sincerely thank the AC and all reviewers' constructive feedback.**

---

### Decision · Program_Chairs · 2025-09-17

**Decision:**

Accept (poster)

**Comment:**

The paper studies a very important topic - reducing memory consumption for the edge LLM inference.
The reviewers engaged in fruitful discussion, with most of their concerns being addressed. The final scores are somewhat mixed (5,4,4,3), but with the majority of reviewers leaning towards acceptance.
I have read the paper carefully, and while I agree with some of the concerns raised by the reviewer aP8N, I think that the NeurIPS community will benefit from this paper being accepted. I recommend to add carefull explanantions in the final version to address aP8N's and other reviwers concerns.